# PixCLIP: Towards Fine-grained Vision-Language Understanding via Any-granularity Pixel-Text Alignment

Yicheng Xiao [* 1 2]   Yu Chen [* 1 2]   Haoxuan Ma [* 3]   Jiale Hong [4]   Caorui Li [5]   Lingxiang Wu [1]   Haiyun Guo [1]
Jinqiao Wang [1]

## Abstract

While CLIP has achieved strong performance across vision–language tasks, fine-grained image–text alignment remains challenging. Recent efforts improve textual granularity by leveraging long, detailed descriptions and replacing CLIP's text encoder with LLM, but often overlook the visual-side bottleneck: achieving finer alignment requires region- and pixel-level visual grounding. To address it, we propose PixCLIP, a framework that jointly enhances both sides by accommodating visual prompt regions and long-form text within a unified training objective. Firstly, to support training at this granularity, we develop an automated annotation pipeline that produces long-form descriptions with pixel-level localization, and use it to construct LongGRIT, a large-scale dataset with 1.5M samples. Furthermore, we introduce a three-branch pixel–text alignment framework that aligns image regions with corresponding textual descriptions across multiple granularities. Experiments show that PixCLIP achieves state-of-the-art performance on pixel- and region-level alignment tasks while preserving strong results on standard global image–text retrieval benchmarks, even with arbitrarily shaped region prompts and long texts. Our code is available at https://github.com/StuHude/PixCLIP.

## 1. Introduction

Contrastive Language-Image Pre-training (CLIP) (Radford et al., 2021) learns transferable visual and textual represen-

---

[1]Institute of Automation, Chinese Academy of Sciences, Beijing, China [2]University of Chinese Academy of Sciences School of Artificial Intelligence, Beijing, China [3]Nanjing University ,Nanjing, China [4]Shanghai Jiao Tong University ,Shanghai, China [5]Southeast University ,Nanjing, China. Correspondence to: Haiyun Guo <haiyun.guo@nlpr.ia.ac.cn >.

*Proceedings of the $43^{rd}$ International Conference on Machine Learning*, Seoul, South Korea. PMLR 306, 2026. Copyright 2026 by the author(s).

tations by aligning images and texts in a shared semantic space, which has significantly advanced tasks such as open-world recognition (Chen et al., 2023; Zhou et al., 2022; Lan et al., 2024), cross-modal retrieval (Huang et al., 2026; Saito et al., 2023; Zhang et al., 2024), and multimodal large language models (MLLMs) (Liu et al., 2023; Yang et al., 2024b). However, many real-world applications demand *fine-grained* vision–language understanding (Hao et al., 2025). For instance, users may query an image with long compositional descriptions that specify multiple constraints (attributes, spatial relations, and explicit exclusions) over an *irregular* region of interest. In such cases, aligning a whole image with a short caption is often insufficient: background content can dominate the embedding, and coarse region proxies (e.g., boxes) cannot precisely isolate the intended pixels. This motivates pixel-level region representations, i.e., embeddings that are explicitly grounded to arbitrary masks, so that the model can both suppress irrelevant background and produce *retrievable* region embeddings aligned with rich textual descriptions.

Recent works have improved fine-grained alignment by modeling long, compositional texts, with LLM-based text encoders such as LLM2CLIP (Huang et al., 2026). However, a key bottleneck remains on the visual side: most methods still operate on global image embeddings and cannot produce *contrastive, retrievable region representations for arbitrarily shaped pixel prompts*. This limitation makes it hard to isolate complex local semantics under rich textual constraints and restricts their applicability to pixel- and region-level understanding and retrieval tasks. In parallel, region-aware CLIP variants (e.g., ReCLIP (Subramanian et al., 2022), CLOC (Chen et al., 2025), Alpha-CLIP (Sun et al., 2024), MaskCLIP (Zhou et al., 2022), and MTA-CLIP (Das et al., 2024)) introduce local region representations, but are typically limited to bounding boxes and short texts, lacking a unified mechanism to align **arbitrary masks** with **long compositional textual descriptions**. Promptable multimodal large language models (Wang et al., 2025; Wu et al., 2024; You et al., 2023) can accept points, boxes, or masks for grounding and segmentation, yet are not designed to output indexable contrastive region embeddings. Therefore, there is a strong need for a CLIP-style framework that

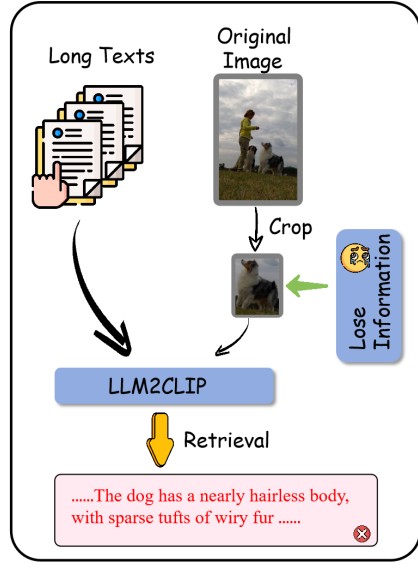 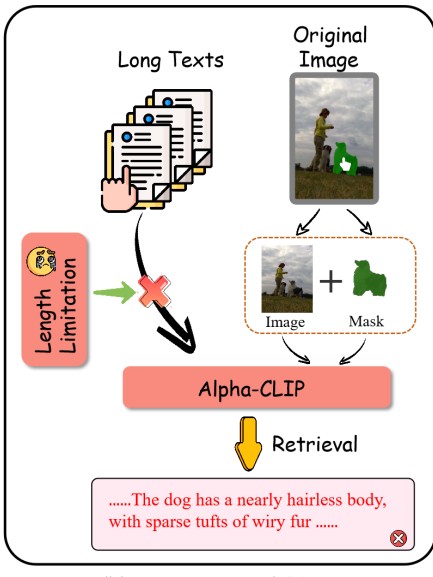 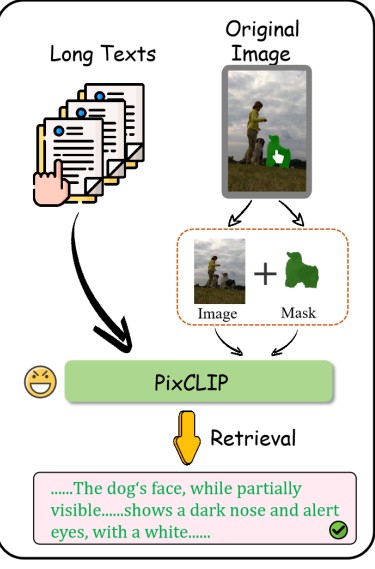

| (a) Long-text CLIP variant | (b) region-aware CLIP variant | (c) PixCLIP (Ours) |

*Figure 1.* As shown on the (a), prior methods fail on fine-grained visual-text alignment tasks due to inability to accept mask input, while existing methods, which support visual prompts, struggle to handle long texts due to token limitations, as shown in (b). Our method succeeds in achieving fine-grained visual-text alignment.

simultaneously supports arbitrarily shaped pixel prompts and arbitrarily long text, while maintaining contrastive and retrievable representations.

However, achieving this objective poses significant challenges. First, there is currently no suitable training data. Existing mask–text datasets, such as GRiT (Peng et al., 2024) and FERRET (You et al., 2023), largely provide phrase-level descriptions and do not offer large-scale mask–text pairs needed for robust alignment between arbitrarily shaped pixel regions and long compositional texts. To fill this gap, we construct **LongGRIT**, a dataset of 1.5M mask–long-text samples. The mask as well as long text caption of each sample is automatically generated and then verified with pixel-level grounding by multiple state-of-the-art multimodal LLMs, yielding a scalable training resource for learning pixel-level, long-text vision–language alignment.

Second, after constructing LongGRIT, we consider a straightforward baseline: directly performing contrastive learning between mask regions and long captions. However, this naive training shows significantly degraded performance across downstream tasks. Guided by our empirical analysis and consistent with observations in prior work (Sun et al., 2024), we hypothesize that the mask–long-text objective is substantially harder and yields noisier effective supervision, making optimization brittle. Specifically, region embeddings can be dominated by background leakage, and local representations may drift from the global semantics learned by the model. To stabilize training and achieve multi-granularity alignment, we introduce a **three-branch framework**. (1) **Contrastive Mask-Text Align-**

**ment** performs the core region–text contrastive learning, aligning mask-guided region features with an LLM-based text encoder. (2) **Fine-Grained Cropping Alignment** enforces consistency between mask-cropped region features and mask-aware features to suppress background leakage. (3) **Local-Global Representation Enhancement** refines the mask embedding by incorporating both local region cues and global context, improving consistency between local and global semantics. Putting these components together, we obtain **PixCLIP**, a CLIP-style model that supports flexible vision–language alignment from global images to arbitrarily shaped pixel regions (Figure 1).

Our main contributions are as follows:

1. We construct **LongGRIT**, a large-scale dataset of **1.5M** mask-long-text pairs, automatically generated and verified with pixel-level grounding by multiple state-of-the-art MLLMs, filling the data gap for learning contrastive alignment between *arbitrarily shaped regions* and *long compositional texts*.

2. Motivated by the instability of naive mask-long-text contrastive training, we propose a **three-branch** alignment framework. Beyond the core **Mask-Text Contrastive Alignment**, we introduce **Fine-Grained Cropping Alignment** to suppress background leakage and **Local-Global Representation Enhancement** to improve consistency between region and global semantics, enabling robust alignment across granularities.

3. Building on LongGRIT and the proposed framework, **PixCLIP** supports *arbitrary region prompts* together

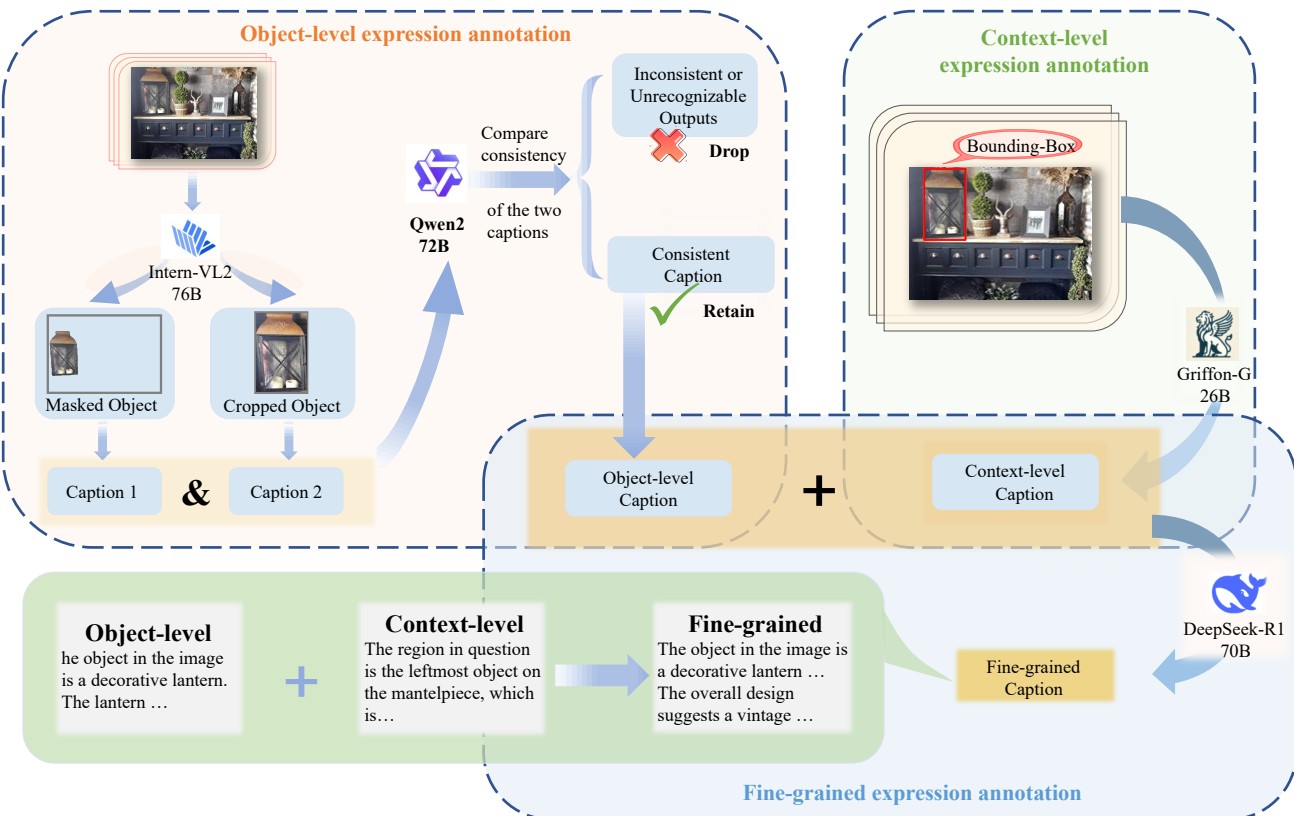

*Figure 2.* The pipeline of our data generation method. Using multiple MLLMs, we generate object captions and in-context captions. They are then checked and merged into fine-grained captions for use in further training stage.

with *arbitrary-length compositional text*, while producing *retrievable contrastive region embeddings*. It achieves state-of-the-art performance on region-level benchmarks (RefCOCO (Lin et al., 2015), Ref-SAV (Yuan et al., 2025), Instance-COCO (Lin et al., 2015), ImageNet-S (Gao et al., 2022)) and maintains strong results on global image-text retrieval datasets (DOCCI, Urban, Flickr).

## 2. Related Works

### 2.1. Regional Image Representation

To enable CLIP to disentangle regions from the whole image for more targeted processing and understanding, various methods have been explored in the field of segmentation. MaskCLIP (Zhou et al., 2022) and ODISE (Xu et al., 2023) use attention masks to make CLIP focus more on local regions. Another approach is to change the input image by simply cropping or masking the image to leave only the foreground object. ReCLIP (Subramanian et al., 2022) and OvarNet (Chen et al., 2023) crop the original image using bounding box from object proposal network. However, the valuable context information is lost except for using complex post-process proposed in ReCLIP (Subramanian et al.,

2022). Some other approaches prompt the CLIP by modifying the input image, guiding CLIP to focus on the area of interest. For example, Red-Circle (Shtedritski et al., 2023), FGVP (Yang et al., 2023) use a circle or mask contour to tell CLIP where to focus. But the direct modification of images causes a domain gap with CLIP pertaining images. methods such as FG-CLIP (Xie et al., 2025) obtain local embeddings by performing bbox-aligned ROI pooling on image features and training them explicitly. This requires additional operations after feature extraction and still cannot address fine-grained understanding at the mask level. Alpha-CLIP (Sun et al., 2024) incorporates an additional alpha channel,which proved that it is a more potential paradigm, but it still suffers from degraded global representation ability and cannot accept long text, with relatively poor performance.

In recent work, some studies also have begun investigating the fine-grained abilities of MLLM, aiming to push the generalization and alignment abilities of MLLM further down to the pixel level. This is similar to our efforts on foundation models. Kosmos-2 (Peng et al., 2024) and CLOC (Iwata et al., 2014) constructed large amounts of bbox-text pairs to train grounding capability, but their methods ultimately only accept bbox input and cannot reach mask-level granularity. GlaMM (Rasheed et al., 2024) and Osprey (Yuan et al.,

2024) build their own mask-caption datasets and introduce additional structures into MLLM that are independent of the vision encoder, but they overlook the importance and potential of the vision encoder itself. They cannot serve as foundation models for retrieval tasks.

## 2.2. Enabling Long Text for CLIP

It has been widely recognized that the quality of CLIP's text embedding is coarse and limited to only 77 tokens. Many works have attempted to extend the length of CLIP captions and retrain CLIP accordingly. DreamLIP (Zheng et al., 2024) leveraged ShareCaptioner (Chen et al., 2024a) and InstructBLIP to augment 30M captions. LongCLIP processes long text by encoding chunks and aggregating them again. These methods were compromised by splitting captions into multiple shorter segments (Zheng et al., 2024; Fan et al., 2023), or fine-tuning the positional encoding to support longer token inputs (Zhang et al., 2024). Recent work LLM2CLIP (Huang et al., 2026) also explores aligning fine-tuned LLM with vision encoder directly. Our model uses LLM as the text encoder and successfully supports the ability for any-granularity visual representation while supporting long text inputs.

## 2.3. Region-level Image Annotation

CLIP is pretrained on large-scale datasets like LAION-400M (Schuhmann et al., 2021) and LAION-5B (Schuhmann et al., 2022), while fine-grained pixel-level labels are not available due to high manual labor costs. CLOC (Chen et al., 2025) also generates fine-grained text labels via the pseudo-labeling pipeline. However, its data format is limited to box and cannot achieve pixel-level annotation, namely mask.Alpha-CLIP uses SAM (Kirillov et al., 2023) and BLIP (Li et al., 2022) to generate 20M object-level caption.But their local captions are either segmented from the entire image caption or generated by the clip-based captioning model, resulting in:(1) It is difficult to ensure the quality of data; (2) All region caption are only composed of several words, with limited length and insufficient fine-grained information. We proposed a fine-grained region-level image annotation framework, which obtained a large number of fine-grained long texts and region pairs from **multiple perspectives**, by using **multiple state-of-the-art MLLMs** and undergoing **multiple verifications**.

## 3. Method

### 3.1. The Construction of LongGRIT

As illustrated in Fig.2, our pipeline builds upon the GRIT-20M (Peng et al., 2024) dataset and consists of three stages: Object-level expression annotation, Context-level expression annotation, and Fine-grained expression annotation.

Each stage progressively refines the target object's description, from appearance attributes to spatial context, culminating in a detailed referring expression. More details for data are provided in Appendix H.

**Object-level expression annotation.** Initially, we isolate each object using its segmentation mask from GRIT-20M. Both the cropped object and its mask are fed into InternVL2-76B (Chen et al., 2024b) to generate an initial *object-level* caption focused on visual attributes (e.g., shape, color, texture). To ensure semantic consistency, we employ Qwen2-72B (Yang et al., 2024a) to validate the caption against the original image. Any captions flagged as inconsistent (e.g., describing non-existent features) or semantically invalid (e.g., "a red clock" for a blue vase) are discarded, while valid captions are retained as *object-level* annotations.

**Context-level expression annotation.** In this stage, we input the original image and the object's bounding box into Griffon-G-26B (Zhan et al., 2024), a vision-language model specialized in spatial reasoning. The model generates a *context-level* caption that describes the object's location (e.g., "leftmost on the mantelpiece") and its relationship to surrounding objects. This complements the object-level caption by capturing contextual information in the first stage.

**Fine-grained expression annotation.** Finally, we integrate the object-level and context-level captions through DeepSeek-R1-70B (Guo et al., 2025), which synthesizes both inputs into a unified *fine-grained* expression. For example, if the object-level caption states 'a decorative lantern' and the context-level caption notes 'the leftmost object on a mantelpiece', DeepSeek-R1-70B merges these into: 'The object is a decorative lantern with intricate carvings, positioned as the leftmost item on a wooden mantelpiece flanked by framed portraits'.

### 3.2. Model Structure

Our PixCLIP implements subtle structural modifications to the CLIP image encoder to preserve CLIP's prior knowledge. Similar to how an image is encoded by a patch embedding layer in vision transformers (ViTs)(Dosovitskiy et al., 2021), we introduce a paralleled mask patch embedding layer to takes in 2D inputs with one channel. We process the input images and masks separately through respective patch embedding layers, then sum their outputs. Specifically, for the inputs images $I$ and Masks $M$, we have the feature $F$:

$$\mathbf{F} = E_n(Conv_I(I) + Conv_M(M) + P) \qquad (1)$$

where $\mathbf{Conv}_I(\cdot)$ and $\mathbf{Conv}_M(\cdot)$ are the image and mask patch embedding layer. Correspondingly, $E_n$ is the vision encoder and $P$ denotes the positional encoding. The newly added mask embedding layer $\mathbf{Conv}_M(\cdot)$ is initialized to output zeros, ensuring that the model's initial behavior is unaffected prior to fine-tuning.

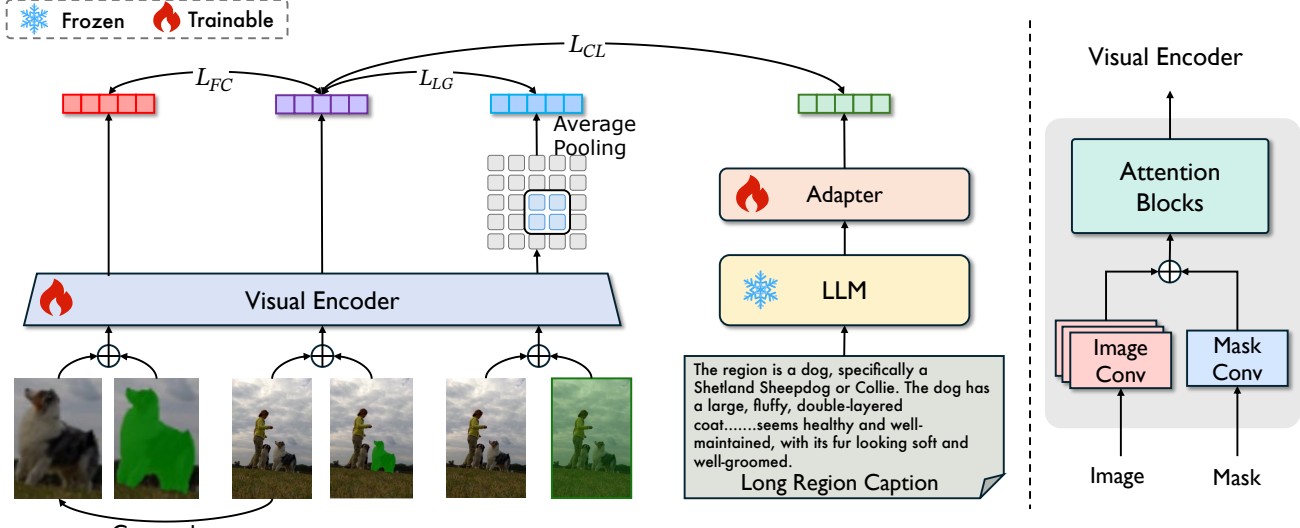

*Figure 3.* The framework of the proposed PixCLIP. Our model structure allows for inheriting weights from existing model and we propose two additional branches: Fine-grained Cropping Alignment $L_{\textbf{FC}}$ and Local-Global Representation Enhancement $L_{\textbf{LG}}$ to enhance the mask-based visual embeddings.

### 3.3. Three-Branch Training Framework

**Overview.** PixCLIP employs three main alignment strategies: global image-text alignment, local mask-text alignment, and multi-scale feature enhancement. As depicted in Fig.3, PixCLIP adopts the architecture likes CLIP, which consists of the vision encoder $V$ and the language encoder $L$, but employs more complex inputs and objectives. We initialized the ViT weights from LLM2CLIP (Huang et al., 2026) and use LLAMA3-8B (Grattafiori et al., 2024) as our text encoder to receive long text input and leverage the open-world knowledge of LLM. Inheriting weights of LLAMA3-8B and the adaptor (a simple projector) from LLM2Vec (BehnamGhader et al., 2024), we first make self-supervised fine-tuning (BehnamGhader et al., 2024) on our text data following its setting and then freezing it as the text tower to reduce training overhead. For training, each input batch consists of $\mathcal{B}$ samples. The input includes an original image $I_i$, a mask $M_i$, and a long text $T_i$ describing the masked region. Additionally, from the original image $I_i$ and mask $M_i$, we generate a *cropped image* $I_i'$ that tightly contains the masked region $M_i$, and a mask $M_i'$ outlining the object within the cropped image $I_i'$, as illustrated in Figure 3. Thus, the full input for sample $i$ is the tuple $(I_i, M_i, T_i, I_i', M_i')$. The long texts $T_i$ are sourced from our LongGRIT dataset.

**Mask-text contrastive learning.** We basically follow the previous CLIP-based settings. For the inputs image and masks $\{I_i, M_i\}$, PixCLIP first encodes them to produce a visual embedding $e_v$ which is focused on the local region referred to by the mask. And LLM encodes the long text to

text embedding $e_t$, the cosine similarity is calculated as:

$$S(\mathbf{e_v}, \mathbf{e_t}) = \frac{\mathbf{e_v} \cdot \mathbf{e_t}}{\|\mathbf{e_v}\|\|\mathbf{e_t}\|} \tag{2}$$

Model is forced to learn visual and text embedding by maximizing the cosine similarity to the corresponding text and image embeddings, while minimizing the cosine similarity to other non-corresponding ones in the batch:

$$L_{\text{CL}}(e_v, \mathbf{e}_t) = -\frac{1}{2\mathcal{B}} \sum_{i=1}^{\mathcal{B}} \left( \log \frac{\exp(S(\mathbf{e}_v^i, \mathbf{e}_t^i)/\tau)}{\sum_{j=1}^{\mathcal{B}} \exp(S(\mathbf{e}_v^i, \mathbf{e}_t^j)/\tau)} \right.$$
$$\left. + \log \frac{\exp(S(\mathbf{e}_t^i, \mathbf{e}_v^i)/\tau)}{\sum_{j=1}^{\mathcal{B}} \exp(S(\mathbf{e}_t^i, \mathbf{e}_v^j)/\tau)} \right) \tag{3}$$

Following previous work (Sun et al., 2024), we set a 10% ratio to occasionally set the mask to all 1 (representing the full image) to maintain the global embedding ability for the entire image. When it occurs, the text is changed to the whole image caption for global alignment.

**Fine-grained cropping alignment.** Some prior methods extract local features from global image representations, often leading to loss of fine-grained details—particularly for small objects in complex scenes. Therefore, to enhance features that are more focused on the region indicated by the mask, on another branch, we cropped the region corresponding to the mask. The cropped region tightly encloses the mask, making the content within the mask the primary focus of the image. After being enlarged, the embedding $v_c, t_c$ obtained from the cropped image $I_c$ and $M_c$ via the visual encoder

is aligned with the original visual embedding, to force the model to focus more on the visual details themselves:

$$L_{FC} = L_{CL(\mathbf{v}_c, \mathbf{t}_c)} \tag{4}$$

Since we want the main embedding to also retain contextual interaction and boundary information, a full alignment with the embedding from the cropped detailed mask is not appropriate. We can only maintain the positive sample matching relationship. This method can also prevent poor model training, keeping the model from focusing too much on the whole image when a mask is provided as input.

**Local-Global representation enhancement.** A model's ability to understand the whole image and its local regions is complementary, meaning their optimization directions tend to converge. A correct global understanding of an image, in fact, stems precisely from the cumulative accurate understanding of all its local regions. This, in turn, dictates whether the model's contextual comprehension of any arbitrary local area is accurate. Therefore, we utilize a branch to explicitly connect these two aspects, combining them into a joint learning task. This branch enables the model to maintain its full-image capabilities and further explore the deep relationship between comprehensive image understanding and arbitrary local comprehension, ensuring both are optimized synergistically.

Specifically, on another branch, we input the original image $I$ with an all-1 mask $M_1$. From the global dense representation, we extract local representations once more. Specifically, regional visual representations $f'_v$ are extracted by pooling visual dense features $\{f^i_v\}_{i \leq \mathcal{P}}$ according to the region occupied by the mask $M$, using Average Pooling:

$$f'_v = proj \left( \text{AvgPool} \left( \{ f^i_v \mid p_i \in \mathcal{M} \cap \mathcal{P}, \frac{|\mathcal{M} \cap \mathcal{P}|}{|\mathcal{P}|} > \rho \} \right) \right) \tag{5}$$

$$L_{LG} = L_{CL(f'_v, f_v)} \tag{6}$$

Then the self-supervision process can continuously update and improve itself.

Finally the total loss is calculated as:

$$L = L_{CL} + \alpha L_{FC} + \beta L_{LG} \tag{7}$$

Noted that throughout the entire training process, short texts are also included and trained together with long texts, in order to enhance the model's robustness.

## 4. Experiment

### 4.1. Experimental Setting

Our experiments evaluated PixCLIP's metrics across multiple dimensions. For evaluation, we used ViT-B/16 models.

Our models were trained on the LongGRIT dataset we made. The batch size was 1024 for ViT-B/16, the $\alpha$ and $\beta$ is set to 0.25. The entire training was completed on 8 H20 100G, totaling 8 epochs.

We compared our approach to previous state-of-the-art (SOTA) models on both pixel-level regional tasks and traditional image-level tasks. For regional tasks, we selected previous works that focused on adapting CLIP for specific areas, including MaskCLIP (Zhou et al., 2022), MaskAdapt-edCLIP (Liang et al., 2023), Red Circle (Shtedritski et al., 2023), Alpha-CLIP (Sun et al., 2024), and so on. We were unable to compare our method with CLOC (Chen et al., 2025) as it has not yet open-sourced its model. In traditional image-text retrieval, our method showed strong performance on both short- and long-text datasets, outperforming models specifically designed for long-text retrieval, such as LongCLIP (Zhang et al., 2024), FG-CLIP(Xie et al., 2025), SigLIP 2(Tschannen et al., 2025) and LLM2CLIP (Huang et al., 2026).

### 4.2. PixCLIP in Region Recognition

**Zero-shot region classification.** We firstly evaluate our model in ImageNet-S (Gao et al., 2022) dataset for zero-shot classification analysis, which comprises 919 classes with semantic segmentation annotations selected from ImageNet-1k. The result is shown as Table.1:

For scenarios that need to crop or mask objects in images (Subramanian et al., 2022; Chen et al., 2023), previous work (Sun et al., 2024) has conducted classification tests for in such scenarios using the validation set of the Instance-COCO (Lin et al., 2015) dataset, which consists of 80 classes. We followed the setting to crop objects using ground-truth bounding boxes and enlarged them by 1.5 times (referred to as coco crop). The experimental results are as Table.3. Importantly, CLIP and LLM2CLIP take the cropped images directly as input.

**Zero shot REC(referring expression comprehension).** For the REC task, we conducted experiments on Ref-COCO(Lin et al., 2015), RefCOCO+(Lin et al., 2015), and RefCOCOg(Lin et al., 2015). Following the setup of Alpha CLIP, we used object proposals predicted by a pretrained detector (Yu et al., 2018) and employed SAM to obtain masks for each proposal. On this task, we significantly outperformed the previous SOTA.

### 4.3. PixCLIP in Multi-modal Retrieval.

As a CLIP-like multimodal foundation model, we meticulously evaluated PixCLIP's performance on cross-modal retrieval tasks. This evaluation included multiple traditional image-text retrieval benchmarks to assess our model's retention of holistic image understanding, alongside our newly

*Table 1.* Accuracy comparison of Zero-shot classification results on ImageNet-S. All methods use ViT-B/16.

| Methods | Input type | Top1 | Top5 |
|---------|-----------|------|------|
| CLIP(Radford et al., 2021) | Crop Image | 66.48% | 88.90% |
| MaskedAdaptedCLIP(Liang et al., 2023) | Crop Image | 57.86% | 79.12% |
| Red Circle(Shtedritski et al., 2023) | Visual Prompt | 65.37% | 88.68% |
| MaskCLIP(Zhou et al., 2022) | Visual Prompt | 67.86% | 89.40% |
| Alpha-CLIP (Sun et al., 2024) | Visual Prompt | 68.89% | 90.51% |
| **PixCLIP** | **Visual Prompt** | **69.57%** | **91.17%** |

*Table 2.* Performance on RefCOCO/+/g Datasets. All models use ViT-B/16.

| Methods | Prompt type | RefCOCO Val | RefCOCO Test | RefCOCO+ Val | RefCOCO+ Test | RefCOCOg |
|---------|-------------|-------------|--------------|--------------|---------------|----------|
| CPT(Yao et al., 2022) | Color | 32.2 | 36.1 | 31.9 | 35.2 | 36.7 |
| ReCLIP(Subramanian et al., 2022) | Bounding-box | 45.8 | 46.1 | 47.9 | 50.1 | 59.3 |
| Red Circle(Shtedritski et al., 2023) | Color + Mask | 49.8 | 58.6 | 55.3 | 63.9 | 59.4 |
| Alpha-CLIP (Sun et al., 2024) | Mask | 55.7 | 61.1 | 55.6 | 62.7 | 61.2 |
| **PixCLIP** | Mask | **59.9** | **69.9** | **59.1** | **68.5** | **61.8** |

*Table 3.* Accuracy comparison on **Instance-COCO**. All methods use ViT-B/16.

| Methods | 0-shot Classification Top1 | 0-shot Classification Top5 |
|---------|------|------|
| CLIP (Radford et al., 2021) | 64.21% | 86.69% |
| Alpha-CLIP (Sun et al., 2024) | 71.08% | 88.90% |
| LLM2CLIP (Huang et al., 2026) | 70.16% | 87.40% |
| **PixCLIP** | **76.68%** | **90.56%** |

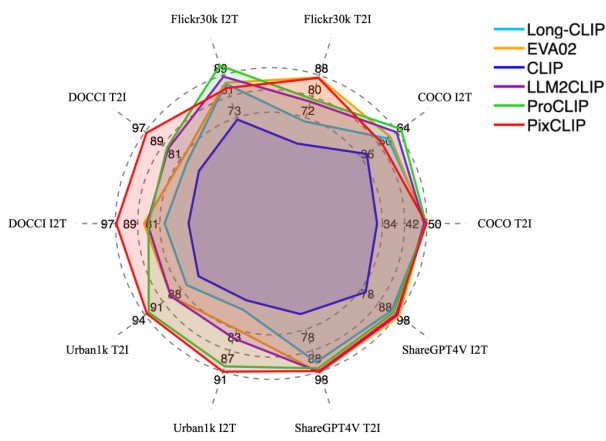

*Figure 4.* Retrieval comparison with previous models in traditional Image-Text benchmarks.

extended pixel-level benchmark: Mask and Text Retrieval. Experimental results clearly show PixCLIP delivers state-of-the-art (SOTA) performance, whether dealing with full images or localized regions, and whether processing concise or extended text descriptions.

**Traditional Retrieval with Full-Image and Text.** For short-text retrieval, we used the MSCOCO (Lin et al., 2015) 5K test set and the Flickr (Young et al., 2014) 1K test set. For long-text retrieval, we employed datasets from Long-CLIP, including a 1K subset of ShareGPT4V (Chen et al., 2024a), the Urban1K (Zhang et al., 2024) dataset, and the DOCCI (Onoe et al., 2024) dataset. The ShareGPT4V-1M dataset consists of captions generated using GPT-4V and ShareCaptioner. Urban1K includes captions for 1,000 urban scene images, each richly annotated with detailed descriptions. DOCCI contains 1.5K high-resolution images with human-annotated captions and was used for retrieval evaluation. Results are shown as Table.4 and Fig.4.

**Pixel-level Retrieval with Mask and Text.** Ref-SAV (Yuan et al., 2025) is a dataset originally intended for video referring segmentation. Ref-SAV includes 70,000 masklets

and corresponding long captions, with an average caption length of over 100 words, providing fine-grained details. Although initially designed for video referring segmentation, we found it particularly well-suited for evaluating our model's performance. We extracted 1,000 mask-text samples for retrieval and compared our work with the previous SOTA work Alpha-CLIP. Our work significantly outperforms Alpha-CLIP in performance.

### 4.4. Ablation Experiment

The PixCLIP training objective consists of three components: $L_{CL}, L_{FC}, L_{LG}$. We conducted the ablation study on these components across both **Pixel-level tasks** (REC

*Table 4.* Retrieval performance comparison on existing datasets.

| Methods | Flickr30k | | COCO | | ShareGPT4V | | Urban-1k | | DOCCI | | Average | |
|---|---|---|---|---|---|---|---|---|---|---|---|---|
| | I2T | T2I | I2T | T2I | I2T | T2I | I2T | T2I | I2T | T2I | I2T | T2I |
| ALIGN | 80.6 | 62.2 | 52.0 | 43.2 | 75.9 | 80.6 | 62.2 | 59.1 | 59.7 | 62.1 | 66.1 | 61.4 |
| BLIP | 80.6 | 74.1 | 61.7 | 48.5 | 65.8 | 74.3 | 45.5 | 48.5 | 50.5 | 53.5 | 60.8 | 59.8 |
| Jina-CLIP | 80.6 | 67.4 | 55.6 | 41.1 | - | - | 87.7 | 88.0 | 78.7 | 80.0 | 75.7 | 69.1 |
| Long-CLIP | 85.8 | 70.6 | 56.9 | 40.9 | 94.8 | 93.5 | 79.1 | 79.1 | 63.1 | 71.4 | 75.9 | 71.1 |
| CLIP | 82.3 | 62.2 | 52.4 | 33.1 | 84.5 | 79.8 | 67.5 | 53.1 | 60.7 | 57.1 | 69.5 | 57.1 |
| EVA02 | 86.2 | 71.5 | 58.7 | 42.1 | 90.5 | 85.5 | 67.0 | 60.8 | 67.7 | 68.0 | 74.0 | 65.6 |
| FG-CLIP | 84.9 | 83.6 | 64.1 | 45.4 | 96.7 | 94.9 | **92.1** | 93.2 | 95.5 | **96.4** | 86.6 | 82.7 |
| SigLIP2 | 81.2 | 83.4 | **65.1** | 48.7 | 90.2 | 87.2 | 75.7 | 74.5 | 77.0 | 78.9 | 77.8 | 74.5 |
| LLM2CLIP | **88.5** | 78.0 | 63.6 | 49.8 | **98.0** | **98.1** | 84.7 | 89.7 | 85.5 | 86.8 | 84.1 | 80.5 |
| **PixCLIP** | 84.2 | **87.0** | 64.4 | **50.2** | 97.4 | 97.7 | 91.0 | **93.7** | 96.7 | 96.4 | **86.7** | **85.0** |

*Table 5.* Accuracy comparison on Zero-shot mask-to-text (M2T) and text-to-mask (T2M) results on Ref-SAV.

| Methods | M2T | | | T2M | | |
|---|---|---|---|---|---|---|
| | Recall@1 | Recall@5 | Recall@10 | Recall@1 | Recall@5 | Recall@10 |
| Alpha CLIP(Sun et al., 2024) | 28.3% | 50.4% | 59.9% | 27.5% | 46.9% | 55.7% |
| **PixCLIP** | **47.3%** | **66.4%** | **73.4%** | **47.9%** | **66.8%** | **74.1%** |

*Table 6.* The ablation results on both pixel-level and image-level tasks.

| Models | Pixel-level Tasks | | | Image-level Tasks | |
|---|---|---|---|---|---|
| | RefCOCO Val | Ref-SAV M2T | Ref-SAV T2M | Urban1k I2T | Urban1k T2I |
| LLM2CLIP(Huang et al., 2026) | - | - | - | 84.7% | 89.7% |
| $L_{CL}$ | 51.144% | 47.0% | 46.9% | 87.1% | 92.4% |
| $L_{CL} + L_{LG}$ | 59.041% | **47.4%** | 47.3% | 88.9% | 93.2% |
| $L_{CL} + L_{LG} + L_{FC}$ | **59.926%** | 47.3% | **47.9%** | **91.0%** | **93.7%** |

*Table 7.* The ablation results on mask embedding types.

| Mask Embedding | DOCCI I2T | DOCCI T2I | Instance-COCO |
|---|---|---|---|
| Add (Original) | **96.7** | **96.4** | **76.7** |
| Concat | 86.6 | 89.9 | 72.9 |

*Table 8.* The ablation on full-image ratio.

| Full-image Ratio | COCO | | Instance-COCO |
|---|---|---|---|
| | I2T | T2I | |
| 0.1 | **64.4** | **50.2** | **76.7** |
| 0.2 | **64.4** | 50.1 | 76.4 |
| 0.3 | 64.3 | 49.8 | 75.9 |

and mask-text retrieval on Ref-SAV) and **Image-level tasks** (Urban1k, COCO and DOCCI).

Our experiments show that simply using direct alignment between mask-based visual features and text is inefficient, leading to poor performance and degradation on whole-image inputs. We hypothesize this is because, while the LLM provides rich text features, the limited amount of local data makes it challenging for the model to learn robust representations that work for both local and global visual inputs. The $L_{FC}, L_{LG}$ successfully resolves this issue and significantly boosts performance. This result precisely confirms the hypothesis we outlined in our method: for strong foundation model, local and global representations can be jointly optimized.

We further conducted in-depth ablation studies on the specific training settings. First, we ablated different ways of incorporating visual prompt embeddings, namely add and concat. As shown in Table.7, the results indicate that add clearly outperforms concat. Second, we ablated the proportion of full-image data used during training. As shown in Table.8, the results demonstrate that appropriately incorporating full-image data with a ratio of 0.1 achieves the best performance.

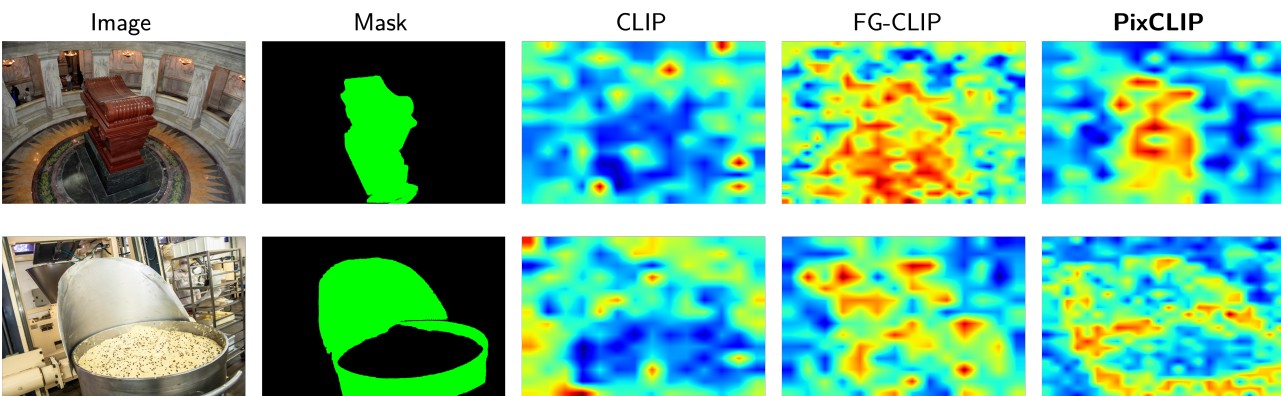

*Figure 5.* Qualitative text-image embedding similarity maps on two challenging examples (top: "an opened scroll on a dark wooden stand over black marble"; bottom: "a silver mixing bowl full of chocolate-chip batter with a metallic rim"). PixCLIP shows stronger fine-grained phrase-to-region alignment and more object-focused responses than CLIP and FG-CLIP.

## 5. Quantitative Results

Following the previous works, we visualized our model's fine-grained responsiveness to text when given a whole image as input. We computed an attention map by calculating the similarity between the feature map from our model's last layer and the text features. As shown in Fig.5, given two identical textual descriptions, PixCLIP shows stronger fine-grained phrase-to-region alignment and more object-focused responses than CLIP and FG-CLIP.

More visualization results are in Appendix.E. We conducted a thorough study to present the visualization results of attention for the masked embeddings and for the text on our PixCLIP.

## 6. Conclusion

We propose PixCLIP, a model capable of accepting arbitrary local input on the image side indicated by a mask and processing text input of arbitrary length on the text side, to achieve deeper, multi-level image-text alignment. PixCLIP achieves SOTA performance whether it receives local pixel-level inputs or full-image inputs, and whether it is paired with short prompts or document-level text. It showes that, after addressing the challenges of data scarcity and framework unification, PixCLIP successfully bridges multiple levels of visual-textual granularity within a single model. We hope PixCLIP will serve as a strong baseline for future works.

## Impact Statement

This paper presents PixCLIP, a CLIP-style framework for fine-grained vision–language understanding that aligns arbitrarily shaped pixel-level regions with long, compositional texts, together with LongGRIT, a large-scale mask–long-text dataset generated via an automated multi-MLLM annotation and verification pipeline. If deployed responsi-

bly, PixCLIP can benefit applications requiring controllable region-level interaction, such as accessibility tools, region-based visual search and content organization, and interactive analysis systems that need precise grounding beyond whole-image matching. Improving region–text retrieval may also reduce reliance on generative outputs when embedding-based search suffices, potentially lowering latency and energy in retrieval-centric deployments. At the same time, stronger region-level alignment can increase capability in harmful settings, such as enabling more targeted surveillance or sensitive attribute retrieval from localized regions. PixCLIP may also inherit biases from pretrained backbones and data, leading to uneven performance across contexts, and users may over-trust high-confidence retrieval results despite residual spurious correlations or failures under distribution shift. LongGRIT is automatically generated; while cross-model verification improves reliability, annotation errors and implicit biases may persist. LongGRIT does not contain human faces or personally identifiable information, and its data sources are licensed appropriately. We position PixCLIP as a research model rather than an autonomous system for high-stakes decisions. For deployment, we recommend standard safeguards including dataset documentation and auditing, bias evaluation, access controls for sensitive-use scenarios, and human oversight in consequential settings. To further ensure a positive social impact, the data we use does not involve any personal privacy issues or risks.

## Acknowledgements

This work was supported by the New Generation Artificial Intelligence-National Science and Technology Major Project (2025ZD0123404) and by the National Natural Science Foundation of China (62276260) . We also thank the anonymous reviewers and the area chair for their constructive feedback during the review process.

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

# A. Related Works

## A.1. Pixel-Level Visual Representation

To enable CLIP to disentangle regions from the whole image for more targeted processing and understanding, various methods have been explored in the field of segmentation and fine-grained understanding. For the earliest methods in the segmentation domain: MaskCLIP uses a 1x1 convolution layer to extract CLIP's final 2D features to obtain semantic information for different regions and uses attention masks to make CLIP focus more on local regions. RegionCLIP generates region box-text pairs for local regions and fine-tunes the CLIP model for box-level recognition. MaskAdapted-CLIP generates mask-text pairs for local masks through a pseudo-labeling process and fine-tunes the CLIP model to make it more adaptable to masked images. These methods suffer from two obvious drawbacks: they are designed with task-specific architectures and trained on corresponding segmentation datasets, which results in very poor generalization; secondly, after enhancing local representation ability, they inevitably impair the global representation capability and cannot maintain the original ability of processing the whole image as input.

Another approach is to change the input image by simply modifying the image to leave only the foreground object, such as cropping or masking the specified region, or using Gaussian blur and blur — ReCLIP and OvarNet follow this idea by cropping the original image using bounding boxes from an object proposal network. However, the valuable context information is lost except when using the complex post-process proposed in ReCLIP. Some other approaches prompt the CLIP by modifying the input image, guiding CLIP to focus on the area of interest. For example, Red-Circle and FGVP use a circle or mask contour to indicate where CLIP should focus. CPT similarly adds colors to distinguish each entity region, mapping RGB to text. But the direct modification of images causes a domain gap with CLIP pretraining images. Alpha-CLIP incorporates an additional alpha channel and proved that it is a more potential paradigm, but it still suffers from degraded global representation ability and cannot accept long text, with relatively poor performance.

In recent work, some studies have begun investigating the fine-grained abilities of MLLM, including local region captioning and VQA tasks, aiming to push the generalization and alignment abilities of MLLM further down to the pixel level. This is similar to our efforts on foundation models. Kosmos-2 and CLOC constructed large amounts of bbox-text pairs to train grounding capability, but their methods ultimately only accept bbox input and cannot reach mask-level granularity. GlaMM and Osprey build their own mask-caption datasets and introduce additional structures into MLLM that are independent of the vision encoder, but they

overlook the importance and potential of the vision encoder itself. They cannot serve as foundation models for retrieval tasks, and they still suffer from short and low-quality text in their datasets.

## A.2. Enabling Long Text for CLIP

It has been widely recognized that the quality of CLIP's text embedding is coarse and limited to only 77 tokens. Many works have attempted to extend the length of CLIP captions and retrain CLIP accordingly. DreamLIP leveraged ShareCaptioner and InstructBLIP to augment 30M captions. LongCLIP processes long text by encoding chunks and aggregating them again. TuLIP and FineLIP extend CLIP by splitting captions into multiple shorter segments, or fine-tuning the positional encoding to support longer token inputs. HiMo-CLIP adds Hierarchical Decomposition after the text embedding to decompose long text into hierarchical sub-semantics. However, these works are still constrained by data and the inherent limitations of the single-transformer architecture, making it difficult to achieve a natural breakthrough in long-text understanding.

With the development of LLMs, recently, works such as LLM2CLIP and ProCLIP have begun to explore aligning fine-tuned LLM with the vision encoder directly, pointing toward a new and more promising direction. However, in practical training, replacing the text encoder with an LLM still faces significant limitations in data and training methodology. As we report in our experiments, simply aligning masks and text after swapping in an LLM does not endow the model with new capabilities; instead, it leads to complete failure.

Ultimately, our model uses LLM as the text encoder and, through high-quality data construction and innovative design, successfully supports any-granularity visual representation while also supporting long-text inputs.

## A.3. Pixel-Level Data Annotation

CLIP is pretrained on large-scale datasets like LAION-400M and LAION-5B, while fine-grained pixel-level labels are not available due to high manual labor costs. CLOC also generates fine-grained text labels via the pseudo-labeling pipeline. However, its data format is limited to box and cannot achieve pixel-level annotation, namely mask.Alpha-CLIP uses SAM and BLIP to generate 20M object-level caption.But their local captions are either segmented from the entire image caption or generated by the clip-based captioning model, resulting in:(1) It is difficult to ensure the quality of data; (2) All region caption are only composed of several words, with limited length and insufficient fine-grained information. We proposed a fine-grained region-level image annotation framework, which obtained a large number of high-quality fine-grained long texts and region pairs from

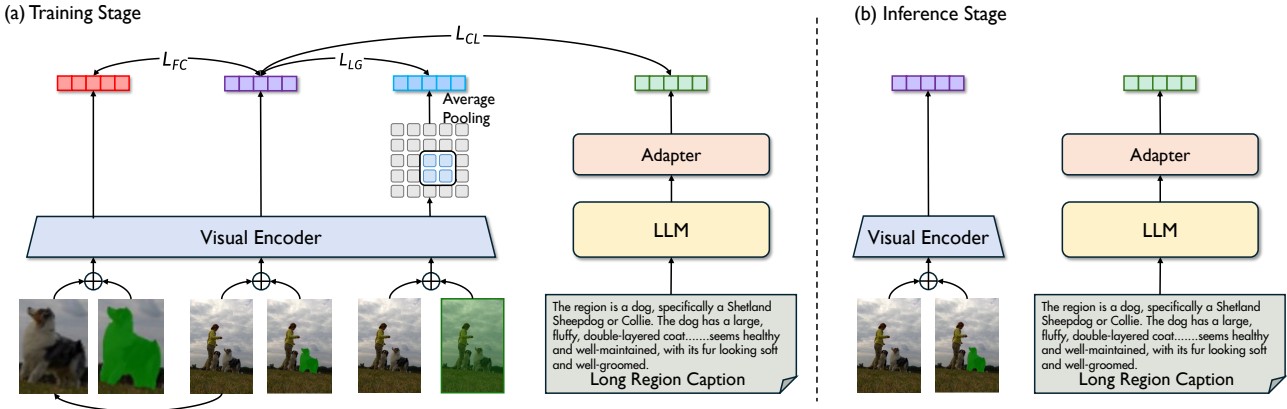

*Figure 6.* The distinction between PixCLIP at training time and inference time.

| Model | MSCOCO | | Flickr30k | | Winoground | | |
|---|---|---|---|---|---|---|---|
| | I2T | T2I | I2T | T2I | T. | I. | G. |
| *Model Architecture: ViT-B/16* | | | | | | | |
| DreamLIP | 53.3 | 41.2 | 82.3 | 66.6 | 26.0 | 10.00 | 7.25 |
| LiT | 30.0 | 16.5 | 54.8 | 38.5 | 24.3 | 6.5 | 4.8 |
| ShareLock | 26.0 | 13.5 | 53.9 | 34.9 | 26.3 | 12.8 | 5.3 |
| CLIP-B | 55.4 | 38.3 | 83.2 | 65.5 | 25.7 | 11.5 | 7.75 |
| SAIL-B-GTE | 48.2 | 37.9 | 76.5 | 63.9 | 31.0 | 11.5 | 9.5 |
| SAIL-B-NV2 | **57.3** | 45.3 | 84.1 | 70.1 | 35.0 | 17.25 | **13.0** |
| PixCLIP | 52.7 | **50.2** | **84.2** | **87.0** | **35.25** | **17.75** | 12.0 |

*Table 9.* Results on complex-reasoning and fine-grained tasks.

**multiple perspectives**, by using **multiple state-of-the-art MLLMs** and undergoing **multiple verifications**.

## B. More Evaluation Results

Due to space limitations in the main manuscript, we further present the experimental results for ViT-L/14 here. We tested and report PixCLIP's performance on Instance-COCO and ImageNet-S.

The results for zero-shot region classification is shown as Table 10 and Table 11. The experiments effectively demonstrate that when provided with a foreground object mask through the alpha channel, our PixCLIP generates visual features that are more focused on the foreground object, leading to better image-level classification compared to the original CLIP and other baseline approaches. Notably, we followed the settings of Alpha-CLIP to enable MaskCLIP to accept task-specific formats. We also visualized and demonstrated PixCLIP's leading performance over previous SOTA work AlphaCLIP on various benchmarks, as shown in Figure.7. Furthermore, we compared the indicators of our method with those of the most advanced approach on the fine-grained analysis benchmark. The results are shown in Table 9.

## C. Detailed Setting

In this section, we list our model architecture and all parameters pertaining to the experimental setup. The detailed parameters are shown in Table 12.

## D. Limitation

PixCLIP still has two significant limitations: **(1)** Although we possess high-quality data, the size of data is insufficient, with only 1.5M samples, hindering effective scaling up. Future approaches might involve more extensive self-supervised training with larger amounts of unlabeled data or constructing more efficient data utilization pipelines. **(2)** When creating the LongGRIT dataset, we did not specifically consider hard negative samples—for instance, having MLLMs intentionally modify a core attribute in a correct description. Future work based on this direction might achieve higher data quality and more efficient data utilization by generating such samples.

| Methods | Input | Imagenet-S | |
|---|---|---|---|
| | | Top1 | Top5 |
| **ViT-B/16** | | | |
| CLIP | C | 66.48% | 88.90% |
| MaskedAdaptedCLIP | C | 57.86% | 79.12% |
| Red Circle | VP | 65.37% | 88.68% |
| MaskCLIP | VP | 67.86% | 89.40% |
| Alpha-CLIP | VP | 68.89% | 90.51% |
| **PixCLIP** | **VP** | **69.57%** | **91.17%** |
| **ViT-L/14** | | | |
| CLIP | C | 73.48% | 91.60% |
| MaskedAdaptedCLIP | C | 73.37% | 92.09% |
| Red Circle | VP | 77.04% | 93.39% |
| MaskCLIP | VP | 77.04% | 93.39% |
| Alpha-CLIP | VP | 77.41% | 94.45% |
| **PixCLIP** | **VP** | **79.42%** | **94.77%** |

*Table 10.* Accuracy on **ImageNet-S**. "C" = Cropped Image. "VP" = Visual Prompt.

| Methods | Zero-shot Classification | |
|---|---|---|
| | Top1 | Top5 |
| **ViT-B/16** | | |
| CLIP | 64.21% | 86.69% |
| Alpha-CLIP | 71.08% | 88.90% |
| LLM2CLIP | 70.16% | 87.40% |
| **PixCLIP** | **76.68%** | **90.56%** |
| **ViT-L/14** | | |
| CLIP | 71.48% | 90.15% |
| Alpha-CLIP | 77.48% | 94.40% |
| LLM2CLIP | xx.xx% | xx.xx% |
| **PixCLIP** | **79.42%** | **94.77%** |

*Table 11.* Accuracy comparison on **Instance-COCO**.

| Parameter | Value |
|---|---|
| Model Structure | EVA02 |
| Backbone | ViT-B/16 |
| Optimizer | AdamW |
| Learning Rate | $10^{-5}$ (Mask Conv) |
| | $10^{-7}$ (Others) |
| $\alpha, \beta$ | 0.25 |
| Warm Up / Epochs | 800 / 8 |
| Batch Size | $128 \times 8$ |
| AMP (bf16) | True |
| Weight Decay | $10^{-2}$ |

*Table 12.* Training parameters.

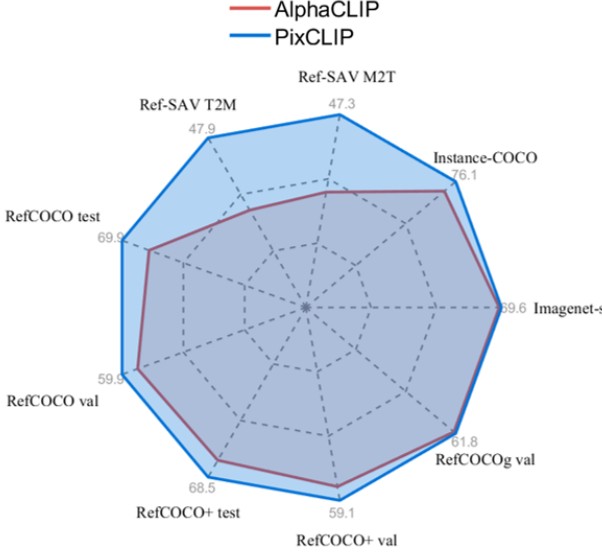

*Figure 7.* Performance comparison between PixCLIP and prior SOTA works.

# E. More Visualization Results

Follow the previous works, we visualize PixCLIP's attention maps to check whether our model pays more attention to user-defined highlighted areas at feature grid space. For a fair comparison, We check the attention map of [CLS] token in the last transformer block in the vision encoder.

Results are shown in Fig.9. This visualization verifies that our model pays more attention to the area to focus on and effectively learns to understand fine-grained semantics.

# F. Inference Stage

During training, we adopt a three-branch framework to train the visual encoder. In the inference stage, the visual encoder only requires the full image and the corresponding mask, without the need for cropping or extracting features from specific regions, as illustrated in Figure 6.

# G. Data Quality

## G.1. Why Different LLMs?

**(1)** We use complementary MLLMs/LLMs with explicit cross-validation. This "division of labor + cross-model verification" mitigates hallucination. In our dataset, only the images are sourced from GRIT, while all textual descriptions are newly generated through our multi-LLM pipeline, thus independent of GRIT data quality. **(2)** Different LLMs have different abilities. Griffon-G is the only-model capable of handling "image + B-box" inputs to describe spatial relations such as "the leftmost vase on the mantelpiece," directly encoding localization cues and addressing questions about incorporating spatial information during training from reviewer.

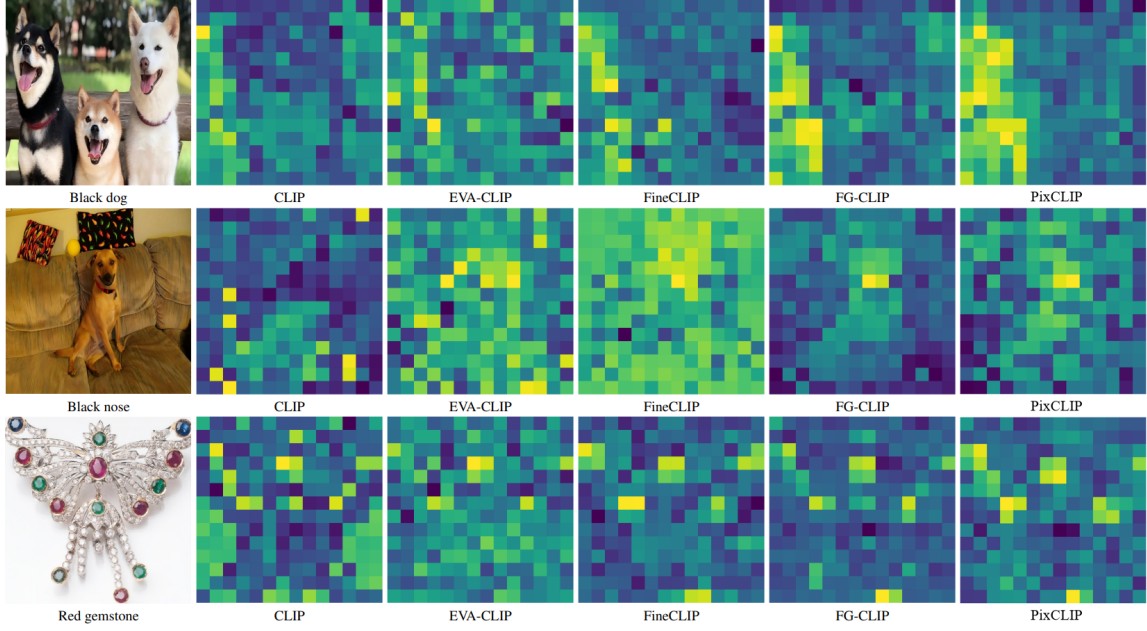

*Figure 8.* Visualized examples of attention map (Full image input).

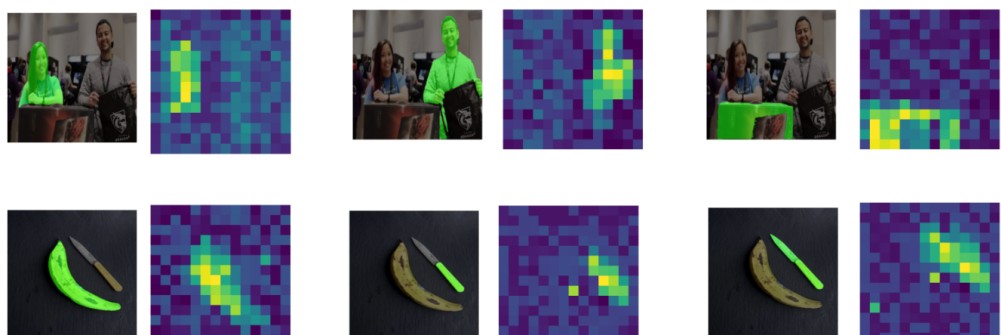

*Figure 9.* Visualized examples of attention map.

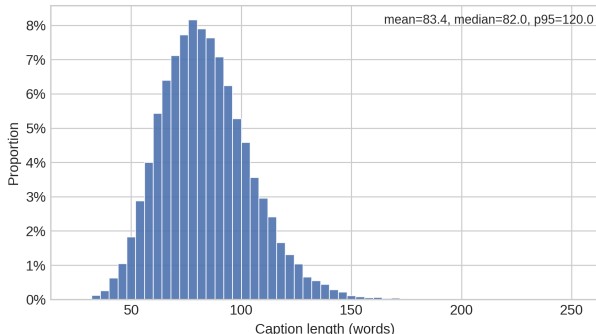

*Figure 10.* Histogram of caption lengths in the LongGRIT dataset

## G.2. Data Detailes

Regarding the details of the LongGRIT dataset we constructed, we provide a histogram of caption word counts, as shown in Fig. 10. The results indicate that captions in our dataset are generally long and contain fine-grained de-

tails. In the first stage of our data construction pipeline, *Object-level expression annotation*, 91.3% of the samples are retained after filtering with the Qwen model.

## G.3. Human Verification

We further conducted a 2,000-sample human audit assessing fine-grained accuracy, achieving over 96% agreement on correctness and spatial coherence.

## G.4. More Examples For Data Generation

We provide some visualized examples of our LongGRIT, shown in Fig. 11 and Fig. 12. Additionally, we showcase the features of our LongGRIT dataset through samples that demonstrate multi-granularity, occlusion reappearance, and both short and long text expressions. These features make LongGRIT also be a challenging and comprehensive Mask-Text Retrieval benchmark.

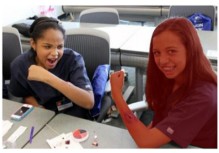

The region features a fake wound or injury, likely used for educational or training purposes, and is located on the arm of a young girl with long, dark hair, wearing a blue uniform with a name tag. The wound looks realistic but is probably made from materials such as latex or silicone to simulate an injury. This kind of prop is often used in medical training, first aid courses, or film and theater productions to create a convincing visual effect. The girl is seated at a table with a red object on it, which appears to be a toy or a piece of equipment, and she is smiling and pointing at the camera, indicating that she is the subject of the photograph. The background is blurred, but it seems to be an indoor setting with other tables and chairs, suggesting a communal or educational environment.

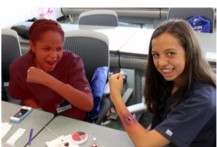

The region features a young girl with a dark complexion, wearing a blue uniform with a name tag. She is seated at a table with a white surface, and her expression is one of surprise or shock. Her hair is pulled back, and she is wearing a headband. The table has a few items on it, including what appears to be a red object that could be a toy or a piece of equipment. The background is blurred, but it seems to be an indoor setting with other tables and chairs, suggesting a communal or educational environment.

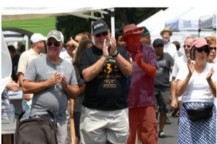

The region features a person calpping hands and the region is located in the top-right part of the image. The person is wearing a hat and a blue shirt, and appears to be participating in a parade or public event. The man in the blue shirt is clapping his hands and appears to be engaged in the event. He has a relaxed posture and is standing in the middle of the crowd, suggesting he is a participant or an attendee at the event. The man's attire is casual, suitable for an outdoor gathering, and his hat suggests it might be a sunny day. The blue shirt he is wearing is a solid color, and the straw hat has a wide brim, which is typical for sun protection.

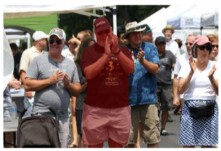

The object in the image is a person clapping their hands. The individual is wearing a black t-shirt with a graphic design, a baseball cap, and sunglasses. They appear to be at an outdoor event, as there are other people visible in the background. The man is the central figure in the image, wearing a black t-shirt with a graphic design and white shorts. He is clapping his hands in front of him, which is a gesture often associated with approval or celebration. His attire and the context suggest he is at a casual, outdoor event. The man's facial expression is not fully visible due to the angle of the photo, but his posture and the action of clapping are the most notable features in this region. The region is located in the center of the image.

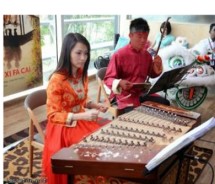

The region contains a man who is playing a traditional Chinese instrument, the erhu, which is a traditional Chinese string instrument often referred to as the "Chinese violin." It consists of a long, narrow neck with two tuning pegs and a small, hexagonal soundbox covered with python skin. The musician plays the erhu using a bow, which is placed between the two strings. The erhu is known for its expressive and haunting sound, and it is commonly used in both traditional Chinese music and contemporary compositions. The man is wearing a red shirt and is seated at a table with the large, His performance attracted a woman and a man in The background to sit and watch him. The setting seems to be indoors with a window in the background showing a view of trees and a building. The region is in the upper right corner of the image.

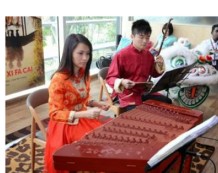

The object in the image is a musical instrument known as a guzheng. The guzheng is a traditional Chinese stringed instrument with a long history, dating back over 2,500 years. It typically has 16 to 26 strings and is played by plucking the strings with the right hand while the left hand presses on the strings to change the pitch. The guzheng is known for its rich, resonant sound and is often used in Chinese classical music and folk music. The region is a musical instrument, specifically a traditional Chinese instrument known as a pipa. It is a four-stringed instrument with a long neck and a round body, which is being played by a person. The instrument is made of wood and has a dark brown color. The strings are white, and the instrument is held with both hands, with a woman's left hand pressing down on the strings to change the pitch and the right hand plucking the strings to produce sound. It is often used in classical and folk music performances. The region is located in the bottom-right part of the image.

*Figure 11.* Visualized examples of our LongGRIT.

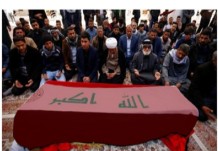

The region features a coffin draped with the national flag of Iraq. The flag features three horizontal stripes of red, white, and black, with the Takbir (الله أكبر, "Allahu Akbar" or "God is the greatest") written in green Kufic script in the center of the white stripe. This type of flag-draped coffin is often used in military or state funerals to honor the deceased. The coffin is covered with a cloth that has a black background and white text, and it is the central focus of the image. The text is in Arabic and is the only text present in the image. The cloth is draped over the object, and the edges of the cloth are visible at the bottom of the image. The object itself is not fully visible, but it appears to be a flat surface, possibly a table or a platform. The cloth has a slight sheen, suggesting it may be made of a material like silk or satin. The text on the cloth is clear and legible, indicating that it is an important element of the image. The region is located in the center of the image.

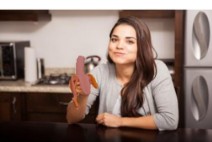

The object in the image is a ripe banana, with a yellow color and a small brown spot. The banana is being held by a woman with her right hand, positioned in front of her and slightly to the left. The woman is wearing a light grey top and has long, dark hair. She is smiling and appears to be in a relaxed and happy mood, standing in a kitchen as the background. The region is located in the center of the image.

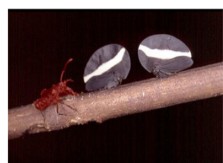

The object in the image is an ant. It is characterized by its small size, brown coloration, and the distinctive body shape of an insect, with a head, thorax, and abdomen. It is positioned on a branch, which is a common behavior for ants as they often traverse branches in search of food or to establish their territory. The ant's antennae are extended, which is typical for ants as they use their antennae for sensing their environment. The background is dark and out of focus, which helps to highlight the ant and the branch it is on. The region is located in the left of the image.

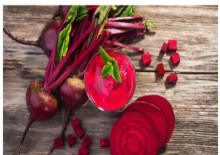

The region in question features a small, red beet slice. It is one of several slices that are scattered around the glass of beet juice. The slice is thin and has a smooth, slightly curved edge. The color is a vibrant red, consistent with the juice's hue, and it appears fresh and crisp. The red beetroot slices connect to the whole beets, juice, and wooden surface, illustrating the vegetable's fresh, natural versatility from raw to prepared. The region is located in the bottom right part of the image.

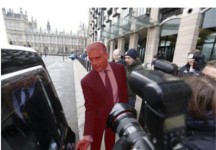

The region is a man, likely a public figure or politician. He is dressed in a dark suit and tie, which is typical attire for formal occasions or professional settings. His posture and the direction of his gaze suggest he is walking purposefully, possibly towards or away from the camera. The man's facial expression is serious, and he appears to be in mid-stride, indicating movement. The background is blurred, which is a common technique in photography to focus attention on the subject. The image does not provide enough detail to determine the man's exact identity or the specific location, but the presence of the camera and the formal attire suggest a setting of significance, possibly related to a public event or a formal engagement. The region is located in the bottom left part of the image.

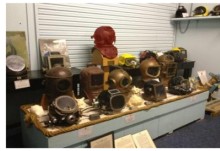

The object in the image is a diving helmet, which appears to be an older model, possibly made of metal, with a round, bulbous shape and several ports or valves on the top. It is displayed among a collection of diving helmets and masks, with various designs, including both traditional and modern shapes, as well as a range of colors from metallic gold to dark brown. The helmets was displayed on a table with many other helmets, many of which had information boards on the front. It indicates that it is in an exhibit or a collection meant for public viewing. The region is located in the bottom left part of the image.

*Figure 12.* More visualized examples of our LongGRIT.

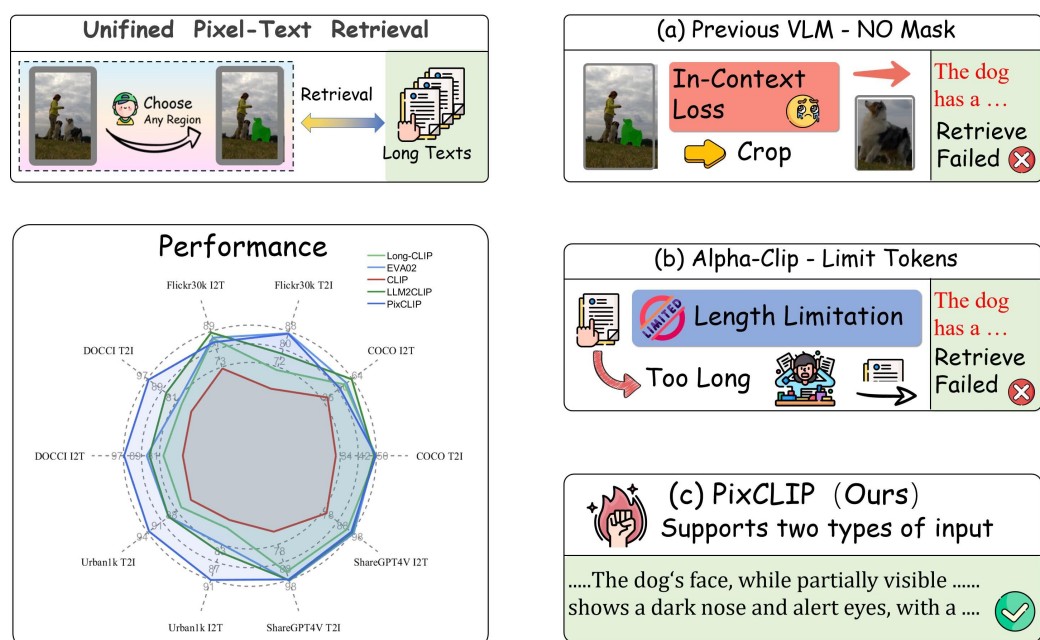

*Figure 13.* PixCLIP: Achieving Fine-grained Visual Language Understanding via Any-granularity Pixel-Text Alignment Learning.

