# OpenReview forum: "PixCLIP: Towards Fine-grained Vision-Language Understanding via Any-granularity Pixel-Text Alignment"
_ICML.cc/2026/Conference — ICML 2026 regular_

### Official Review · Reviewer_QZhF · 2026-02-27

**Soundness:** 2
**Presentation:** 2
**Significance:** 3
**Originality:** 2
**Overall Recommendation:** 2
**Confidence:** 4

**Summary:**

This paper aims to solve the fine-grained vision-language alignment bottleneck of CLIP models, which cannot accurately align arbitrary-shaped pixel regions with long composite texts. To address this issue, the authors construct the LongGRIT dataset and propose the PixCLIP framework with a three-branch training strategy.

**Compliance With Llm Reviewing Policy:**

Affirmed.

**Final Justification:**

I appreciate the authors' rebuttal, but my main concerns remain unresolved, so I maintain my original score. While the rebuttal addressed some points partially, I still find the methodological novelty limited, as the proposed approach relies heavily on weights from LLM2CLIP and LLM2Vec and does not show sufficient structural innovation compared with prior works such as Alpha-CLIP and LLM2CLIP. I am also concerned about the experimental setup and academic rigor. In particular, Figure 5 uses the exact same visualizations as FG-CLIP without referencing or comparing against that method in the main paper. Although some additional results were provided in the rebuttal, a more complete and systematic evaluation is still needed. LongGRIT is presented as a major contribution, but its overall impact is weakened by the lack of a clear open-source plan, which raises reproducibility concerns. The manuscript also contains some typos that should be corrected. Overall, this paper still has room for improvement.

**Key Questions For Authors:**

Please refer to Weaknesses. I would like to increase the score if my concerns are solved.

**Limitations:**

yes

**Strengths And Weaknesses:**

**Strengths**

Based on the segmentation masks of GRIT-20M, LongGRIT dataset adopts a scientific three-stage automatic annotation and cross-validation mechanism with multiple state-of-the-art multimodal large language models (MLLMs), and the manual review of 2000 samples shows an accuracy rate of over 96%, which provides reliable data support for the research.

**Weaknesses**

1. There are a few typos in the paper, which requires strict proofreading to ensure the accuracy and professional presentation.

2. PixCLIP is improved on the basis of Alpha-CLIP, mainly replacing the text encoder with that of LLM2CLIP. This raises concerns about the extent of technical innovation. Moreover, it remains unclear whether the observed performance gains over Alpha-CLIP stem primarily from the LongGRIT dataset or the updated text encoder.

3. Some recent baseline results need to be discussed and compared, such as FG-CLIP [1] and SigLIP 2 [2].  In particular, Figure 5 uses the exact same visualization images as those in the FG-CLIP paper, but FG-CLIP is not discussed at all.

4. The paper employs LLaMA-3-8B as the text encoder, whereas most baseline CLIP-style models use text encoders with fewer than 1 billion parameters. It would strengthen the fairness of the comparison if the authors report results using smaller LLMs (e.g., 1B-scale) to isolate the impact of architectural changes from the model size.

5. In Section 4.2, PixCLIP scales up the cropped target by 1.5 times, while CLIP and LLM2CLIP use the original size. The reason for this experimental configuration is not explained clearly.

[1] FG-CLIP: Fine-Grained Visual and Textual Alignment. ICML 2025.

[2] Siglip 2: Multilingual vision-language encoders with improved semantic understanding, localization, and dense features. arXiv preprint arXiv:2502.14786, 2025.

---

> ### Author Rebuttal · Authors · 2026-03-30
>
> **Q2:** We respectfully clarify that our main contribution is not simply replacing the text encoder.
>
> **First**, our experiments show that directly aligning the visual and text modalities through standard training leads to failure. This motivates our proposed ***three-branch training framework***, which significantly improves performance; this methodological innovation is a key factor behind the gains.
>
> **Second**, the contribution of the dataset lies in its necessity. Without it, we would not have paired mask–longtext data, and therefore could not perform alignment between arbitrary pixel regions and arbitrarily long textual descriptions.
>
> **Third**, the updated text encoder serves as an implementation choice that enables ***"handling long-text inputs"*** and ***"leveraging the prior knowledge of LLMs"***, rather than being our core contribution. In fact, the original CLIP text encoder cannot process inputs longer than 77 tokens.
>
> If the reviewer suspects that the performance improvements mainly come from the updated text encoder, this can be verified empirically. AlphaCLIP avoided discussing the preservation of global image capabilities, as the experiments demonstrated that endowing the model with new local capabilities often leads to a loss of its original global capabilities. If our model’s performance improvement were solely due to replacing the text encoder, it would be impossible for it to outperform llm2clip—which is specifically designed for text-image tasks using an LLM—in terms of global image capabilities. We will include additional experimental evidence and analysis regarding this section in the appendix of the main text.
>
> | Method | COCO |  | ShareGPT4V |  | Urban1K |  | Avg |  |
> | --- | ---: | ---: | ---: | ---: | ---: | ---: | ---: | ---: |
> | &nbsp; | I2T | T2I | I2T | T2I | I2T | T2I | I2T | T2I |
> | CLIP | 52.4 | 33.1 | 84.5 | 79.8 | 67.5 | 53.1 | 68.1 | 55.3 |
> | AlphaCLIP | 49.0 | 33.6 | 72.9 | 69.5 | 58.9 | 54.7 | 60.3 **(-7.8)** | 52.6 **(-2.7)** |
>
>
>
> &nbsp;
>
> **Q3:** The comment is very valuable. We have taken it into account and have supplemented our paper with experimental comparisons against these two models :
> | Methods | Flickr30K I2T | Flickr30K T2I | DCI I2T | DCI T2I | COCO I2T | COCO T2I | Urban1K I2T | Urban1K T2I | DOCCI I2T | DOCCI T2I | ShareGPT4V I2T | ShareGPT4V T2I | Avg. I2T | Avg. T2I |
> | --- | ---: | ---: | ---: | ---: | ---: | ---: | ---: | ---: | ---: | ---: | ---: | ---: | ---: | ---: |
> | FG-CLIP | **84.9** | 83.6 | 61.8 | 60.6 | 64.1 | 45.4 | **92.1** | 93.2 | 95.5 | **96.4** | 96.7 | 94.9 | 82.5 | 79.0 |
> | SigLIP2 | 81.2 | 83.4 | —— | —— | 65.1 | 48.7 | 75.7 | 74.5 | 77.0 | 78.9 | 90.2 | 87.2 | 77.8 | 74.5 |
> | PixCLIP | 84.2 | **87.0** | **62.2** | **69.3** | **64.4** | **50.2** | 91.0 | **93.7** | **96.7** | **96.4** | **97.4** | **97.7** | **86.7** | **85.0** |
>
> **（①.** we found that our originally reported PixCLIP result on COCO I2T was incorrect due to a coding error, which led to an underestimated performance. We have now corrected this issue and updated the values. We have now corrected the values and can ensure that the following results are accurate.
>
>  **②.** We did not report SigLIP2 results on DCI because its text encoder has a length limitation; when evaluated on the long-text data in DCI, the performance is extremely low and therefore not meaningful.**）**
>
> The experimental results show that our PixCLIP achieves superior performance on the both average metrics. More results here https://anonymous.4open.science/r/pixclip-figure-BBE0/pixclip-viz-caption.png
>
> &nbsp;
>
> **Q4:** Following your suggestion, we replaced the text encoder from 8B with Llama-3.2-1B. The experimental results are shown in the figure below:
> | Text Encoder | Flickr I2T | Flickr T2I | COCO I2T | COCO T2I | ShareGPT4V I2T | ShareGPT4V T2I | Urban I2T | Urban T2I | DOCCI I2T | DOCCI T2I | Avg. I2T | Avg. T2I |
> | --- | ---: | ---: | ---: | ---: | ---: | ---: | ---: | ---: | ---: | ---: | ---: | ---: |
> | Llama-3-8B | 84.2 | 87.0 | 64.4 | 50.2 | 97.4 | 97.7 | 91.0 | 93.7 | 96.7 | 96.4 | 86.7 | 85.0 |
> | Llama-3.2-1B | 83.9 | 86.5 | 62.1 | 48.3 | 96.9 | 97.1 | 89.2 | 92.2 | 94.7 | 94.9 | 85.4 | 83.8 |
>
> The experimental results show that after replacing the text encoder with a 1B model, our method still outperforms prior works such as ***LLM2CLIP (Avg: 84.1 & 80.5)*** and ***FG-CLIP (Avg: 82.5 & 79.0)***.
>
> &nbsp;
>
> **Q5:** Reviewer may have misunderstood our implementation. Under this benchmark’s evaluation setting, all models take as input a cropped region that is enlarged by 1.5×. This applies uniformly to LLM2CLIP, CLIP, and our model. In other words, for standard models the input is the bounding box treated as an image, while for pixel-level models the input consists of the bounding box together with a mask that occupies the main region of the image. Without this preprocessing, standard models such as LLM2CLIP would not be able to effectively encode local regions.

---

> > ### Author Rebuttal · Reviewer_QZhF · 2026-04-02
> >
> > The authors have partially addressed my concerns. However, the response to Q2, as well as the provided table, still does not directly answer my question. The current evidence remains insufficient for a clear attribution of the observed gains.

---

> > > ### Author Response · Authors · 2026-04-04
> > >
> > > Thank you for the thoughtful follow-up. We are encouraged that the added evidence on the comparisons with FG-CLIP and SigLIP2, and replacing with a smaller text encoder, has strengthened the empirical support and improved the transparency of the paper.
> > >
> > > We also appreciate that, among all your questions, only Q2 remains unresolved; we now provide a focused and detailed response to Q2 to clarify our contributions. We would like to restate the contributions of this paper:
> > >
> > > &nbsp;
> > >
> > > Our ultimate goal is **to train a foundation model that can align arbitrary local regions with arbitrarily long text.**
> > >
> > > **①** Therefore, at architecture level, we design an additional mask convolution layer on the visual side and replace the text encoder with an LLM. **This is the necessity of the model structure.**
> > >
> > > **②** Then, since there is no Region–long-text dataset available for training this objective, we have to design an automated pipeline to construct our own dataset, LongGRIT. **This is the necessity of LongGRIT**, without it, we simply cannot train a model that aligns regions with long text.
> > >
> > > **③** Next, after modifying the model architecture, we directly train using LongGRIT (without any auxiliary branches), and the results are reported in the ablation study Table 6 in our paper. The performance is already competitive, although still inferior to previous SOTA in some aspects. This demonstrates **the effectiveness of the Model Structure + LongGRIT**.
> > >
> > > To demonstrate the effectiveness of our model design and dataset, we further add experiments where AlphaCLIP’s text encoder is replaced to suit the new task and then trained on LongGRIT:
> > >
> > > | Method | ImageNet-S Acc@1 | Flickr30K I2T | Flickr30K T2I | ShareGPT4V I2T | ShareGPT4V T2I | Instance-COCO Acc@1 |
> > > | --- | ---: | ---: | ---: | ---: | ---: | ---: |
> > > | AlphaCLIP | 68.9 | 70.0 | 72.0 | 72.9 | 69.5 | 71.1 |
> > > | AlphaCLIP + LongGRIT | 69.3 | 82.2 | 82.4 | 93.8 | 94.0 | 74.2 |
> > >
> > > Results show that such a model naturally achieve performance between the original AlphaCLIP and our final PixCLIP, further demonstrating the effectiveness of LongGRIT. Since AlphaCLIP was not trained on high-quality long-text data, it performs much worse on image–text retrieval datasets such as ShareGPT4V.
> > >
> > > To further demonstrate the contribution of our dataset, we also conduct data scaling experiments:
> > >
> > > | Data Scale | ImageNet-S @1 | ImageNet-S @5 | DOCCI I2T | DOCCI T2I | RefCOCO Val | FG-OVD trivial |
> > > | --- | ---: | ---: | ---: | ---: | ---: | ---: |
> > > | 20% | 65.7 | 88.5 | 96.7 | 96.0 | 56.8 | 52.6 |
> > > | 50% | 67.5 | 90.0 | 96.5 | 96.3 | 58.4 | 69.2 |
> > > | 100% | 69.6 | 91.2 | 96.7 | 96.4 | 59.9 | 78.9 |
> > >
> > > **④** Finally, to address the difficulty of training and the insufficient performance, we design two dedicated branches to assist the features from the main branch during training. The LG branch jointly improves global and local capabilities, while the FG branch enlarges local details as visual anchors, as shown in the ablation study, they all boost performance, show **the effectiveness of the training framework**. We again provide the ablation study from our paper:
> > >
> > > | Models | RefCOCO Val | Ref-SAV M2T | Ref-SAV T2M | Urban1k I2T | Urban1k T2I |
> > > | --- | ---: | ---: | ---: | ---: | ---: |
> > > | LLM2CLIP | - | - | - | 84.7% | 89.7% |
> > > | L_CL | 51.144% | 47.0% | 46.9% | 87.1% | 92.4% |
> > > | L_CL + L_LG | 59.041% | **47.4%** | 47.3% | 88.9% | 93.2% |
> > > | L_CL + L_LG + L_FG | **59.926%** | 47.3% | **47.9%** | **91.0%** | **93.7%** |
> > >
> > > &nbsp;
> > >
> > > All components of our paper are organized as above. Each step is designed to achieve the final goal, and has been verified through our attempts and experiments. The logic is clear and consistent. We hope the reviewer can understand the overall reasoning of our work and no longer feel confused.
> > >
> > > &nbsp;
> > >
> > > Sincerely thank again for your careful review, although you may have some confusion. **If there is anything you want us to further address, please reply to us.**

---

### Official Review · Reviewer_BXGd · 2026-03-01

**Soundness:** 4
**Presentation:** 2
**Significance:** 4
**Originality:** 4
**Overall Recommendation:** 5
**Confidence:** 4

**Summary:**

The paper studies fine-grained vision-language understanding, focusing on aligning regional text descriptions with corresponding arbitrarily shaped pixel region. To this end, the authors first construct LongGRIT, a large scale mask-long-text pairs dataset using multiple MLLMs. And then, they present PixCLIP, a framework that supports any-granularity pixel-text alignment via mask-based visual prompting. With three dedicated training objectives, PixCLIP demonstrates SoTA performance on both region-level and global-level tasks.

**Compliance With Llm Reviewing Policy:**

Affirmed.

**Final Justification:**

Rebuttal addressed my concerns. I raise score: **accept**

**Key Questions For Authors:**

>  **I am open to revising my score and would consider raising it if the concerns are clarified.**

**Q1. Comparision between Alpha-CLIP’s visual prompting and PixCLIP’s design**

If the rest of the framework (three losses, dataset, etc) is kept unchanged, how does the performance change when only the visual prompting mechanism is replaced with Alpha-CLIP’s approach? (additive mask prompting *vs* 4-channel inputs)

**Q2. Evaluate on FG-OVD benchmark**

FG-OVD [1] is a benchmark for recognizing regions within given bounding boxes and appears well-suited to evaluate PixCLIP’s capabilities.
(1) Following the evaluation protocol in FG-CLIP [2], compare PixCLIP with FG-CLIP using all-1 mask and ROI pooling.
(2) Additionally, use the provided bounding boxes as prompts $M$ and report the performance.

**Q3. Additional global-level vision-language alignment?**

PixCLIP does not perform *direct* global-level image-text alignment during training (i.e., No use of full-scene caption, only region captions). It would be interesting to understand whether full-scene caption would further improve model over $L_{LG}$. Given the strong results on DOCCI and Urban-1k, I understand that PixCLIP is already good at global-level understanding.

**Q4.  Dataset will be released?**

The manuscript does not state whether LongGRIT will be publicly released. Clarifying the release plan would improve reproducibility and community impact.

---

[1] Bianchi et al. The devil is in the fine-grained details: Evaluating open-vocabulary object detectors for fine-grained understanding. CVPR’24

[2] Xie et al. FG-CLIP: Fine-Grained Visual and Textual Alignment. ICML’25

**Limitations:**

yes

**Strengths And Weaknesses:**

**Strengths:**

**S1.** (*Major*) **Any-granularity vision-language understanding.**

PixCLIP supports both region-level (mask prompts) and image-level (all-1 mask) representations within *unified framework*.

**S2.** (*Major*) **Novelty of Local-Global Representation Enhancement ($L_{LG}$).**

The use of an all-1 mask to define a global representation and form self-supervised alignment with regional features is a novel and effective design.

**S3.** (*Minor*) **Dataset contribution (LongGRIT).**

LongGRIT is a useful dataset, particualry due to its way to integrate of object-level and context level descriptions.

---

**Weakness:**

**W1.** (*Major*) **The notation and illustration of** $L_{FC}$ **is unclear.**

- According to Figure 3, $L_{FC}$ appears to represent a loss between two visual embeddings (red-purple). However, the notation $t_c$ and the description referring to “maintaining only a positive relationship” suggest that it may instead be a loss between a visual embedding and a textual embedding (red-green). This should be clarified.
- Is $t_c$ identical to $e_t$, or $e_v$?
- Are $I_c$ and $M_c$ the same as $I’_i$ and $M’_i$?

**W2.** (*Minor*) **Comparison with Alpha-CLIP**

Alpha-CLIP also uses masks, but its captions are relatively short compared to the long captions in LongGRIT. While LongGRIT itself is a contribution of this work, it is unclear whether the performance gain comes from the framework or the dataset.

---

> ### Author Rebuttal · Authors · 2026-03-31
>
> We thank Reviewer `BXGd` for the encouraging review.  Below we reply.
>
> **W1:** **①** Yes. $L_{FC}$ represents a loss between two visual embeddings. **②** Here, $v_c$ and $t_c$ denote the visual and textual features corresponding to the cropped image, while $e_t$ and $e_v$ refer to the text and visual features in the contrastive learning formulation. The two pairs are consistent; the former can be viewed as instances of the latter in the cropped-data setting. **③** $I_c$ and $M_c$ are indeed the same as $I_i^l$ and $M_i^l$, representing an additional Image+Mask pair in the input. This pair is obtained by cropping and enlarging the original data $I_i$ and $M_i$. We apologize for the confusion and will clarify this in the final version.
>
> &nbsp;
>
> **Q1:** In fact, additive mask prompting and 4-channel inputs are essentially the same: both use an additional convolutional layer to encode the mask input. Since the mask is single-channel, combining it with the RGB image results in a four-channel input. The two are then added to produce the input features before self-attention.
>
> Therefore, the mask embedding implementations of the two approaches are largely identical. The differences mainly lie in details such as whether positional encoding is applied and the specific convolutional parameters. We can also provide ablation studies on different visual prompt embeddings (which were not explored in AlphaCLIP). In fact, we did not initially choose this visual prompt design; rather, it was selected based on empirical comparisons showing it performs best:
>
> | Mask Embedding Type | DOCCI I2T | DOCCI T2I | Instance-COCO |
> | --- | ---: | ---: | ---: |
> | Add (Original) | 96.7 | 96.4 | 76.7 |
> | Concat | 86.6 | 89.9 | 72.9 |
>
> Replacing add with concat, followed by conv before feeding into self-attention, leads to inferior performance.
>
> &nbsp;
>
> **Q2:** This is a very valuable suggestion. Other reviewers have also requested comparisons with FG-CLIP. We have now added extensive experiments comparing our method with recent state-of-the-art models, including FG-CLIP and SigLIP2, across multiple benchmarks. First, we present the requested experiment:
>
> | Method | Backbone | Emb Implementation | FG-OVD hard | FG-OVD medium | FG-OVD easy | FG-OVD trivial |
> | --- | --- | --- | ---: | ---: | ---: | ---: |
> | CLIP | ViT-B/16 | Feature ROI | 12.0 | 23.1 | 22.2 | 58.5 |
> | PixCLIP**①**  | ViT-B/16 | Feature ROI | 36.9 | 48.6 | 57.4 | 68.4 |
> | PixCLIP**②**  | ViT-B/16 | Bbox + Image | 44.2 | 65.2 | 66.8 | 77.9 |
> | FG-CLIP | ViT-B/16 | Feature ROI | **46.1** | 66.6 | 68.7 | **83.4** |
> | PixCLIP**③**  | ViT-B/16 | (Bbox->Mask) + Image | 45.2 | **67.0** | **71.3** | 78.9 |
>
> In the table, ① denotes applying our model under the FG-OVD setting, where we first use an all-1 mask to obtain full-image features and then perform ROI. ② denotes directly using the bbox from FG-OVD as the visual prompt input to our model. ③ denotes attaching an external SAM to first convert the bbox into a mask, which is then used as the visual prompt input (this highlights our model’s advantage in fine-grained modeling, whereas models like FG-CLIP can only take bbox inputs). As shown, our model remains competitive.
>
> On the other hand, we also compared our results with those of FG-CLIP on other traditional fine-grained image-text benchmarks:
>
> | Methods | Flickr30K I2T | Flickr30K T2I | DCI I2T | DCI T2I | COCO I2T | COCO T2I | Urban1K I2T | Urban1K T2I | DOCCI I2T | DOCCI T2I | ShareGPT4V I2T | ShareGPT4V T2I | Avg. I2T | Avg. T2I |
> | --- | ---: | ---: | ---: | ---: | ---: | ---: | ---: | ---: | ---: | ---: | ---: | ---: | ---: | ---: |
> | FG-CLIP | **84.9** | 83.6 | 61.8 | 60.6 | 64.1 | 45.4 | **92.1** | 93.2 | 95.5 | **96.4** | 96.7 | 94.9 | 82.5 | 79.0 |
> | PixCLIP | 84.2 | **87.0** | **62.2** | **69.3** | **64.4** | **50.2** | 91.0 | **93.7** | **96.7** | **96.4** | **97.4** | **97.7** | **86.7** | **85.0** |
>
> More results here: https://anonymous.4open.science/r/pixclip-figure-BBE0/pixclip-viz-caption.png
>
> &nbsp;
>
> **Q3:** This is already included in the current method. As stated in the paper, we set a `10%` ratio to replace the region mask with an all-1 mask and replace the region text with the whole-image caption;  But it is indeed a valuable point, so we have added ablation studies on the parameter choices to illustrate their impact.
> | Full-image Ratio | COCO I2T | COCO T2I | Instance-COCO Classification | FG-OVD trivial |
> | --- | ---: | ---: | ---: | ---: |
> | 0.1 | **64.4** | **50.2** | **76.7** | **78.9** |
> | 0.2 | **64.4** | 50.1 | 76.4 | 77.6 |
> | 0.3 | 64.3 | 49.8 | 75.9 | 74.2 |
>
> However, this does not necessarily indicate the optimal ratio. The current results may be influenced by the fact that our high-quality training data primarily consists of Mask–Text pairs, rather than high-quality Image–Text pairs. We will provide more detailed analysis and additional experiments in the appendix of the final version.
>
> &nbsp;
>
> **Q4:** Yes.

---

> > ### Author Rebuttal · Reviewer_BXGd · 2026-04-02
> >
> > Most of my concerns have been fully resolved, with only a minor question remaining.\
> > I will raise my final score to **Accept**.
> >
> > ---
> >
> > **[W1, Q1, Q2, Q3]** - **Fully resolved.**
> >
> > **[Q2]** - I found the response particularly **insightful**.\
> >  The FG-OVD experiments well demonstrate PixCLIP’s fine-grained regional understanding, and also provide useful insights into how performance evolves across the ① $\rightarrow$ ② $\rightarrow$ ③ settings. I consider setting ③ to be a fair evaluation that aligns well with the proposed method.
> >
> > **[W2]** - This is the only remaining point, but I consider it minor. \
> > I was primarily curious about the comparision with "Alpha-CLIP method + LongGRIT dataset", to assess the dataset’s contribution to performance.
> > I acknowledge that this is beyond the scope of the rebuttal period, and a brief discussion would be helpful.

---

> > > ### Author Response · Authors · 2026-04-02
> > >
> > > Thank you for the thoughtful follow-up. We are encouraged that the added evidence on the Mask embedding Type, FG-OVD eval and Full-Image Ratio has strengthened the empirical support and improved the transparency of the paper. We also appreciate your constructive remaining concerns on the LongGRIT, and we address them below, and we would like to clarify the underlying logic:
> > >
> > > A CLIP-style model consists of both a visual encoder and a text encoder. For AlphaCLIP to align with long-text descriptions, it requires not only a dataset such as LongGRIT (which provides Mask–long-text pairs), but also a text encoder capable of processing long sequences. However, AlphaCLIP adopts the standard CLIP text encoder, which is limited to 77 tokens (i.e., it can only effectively encode the first 77 tokens of a long input). Therefore, directly training AlphaCLIP with LongGRIT would fail to properly model long-text descriptions, leading to collapsed and meaningless results. This also explains why LongGRIT is not only a contribution but a necessity: without it, we cannot train a model capable of aligning arbitrary local regions with arbitrarily long text.
> > >
> > > Alternatively, one could follow the setting of PixCLIP by first replacing the text encoder of AlphaCLIP with an LLM and then training. In this case, the resulting performance would be similar to our ablation results where additional training branches are removed (as reported in Table 6). That is, simply using a single training branch aligned with the text features leads to suboptimal performance. The only difference would lie in the initialization from AlphaCLIP’s pretrained weights.
> > >
> > > | Method | ImageNet-S Acc@1 | Flickr30K I2T | Flickr30K T2I | ShareGPT4V I2T | ShareGPT4V T2I | Instance-COCO Acc@1 |
> > > | --- | ---: | ---: | ---: | ---: | ---: | ---: |
> > > | AlphaCLIP | 68.9 | 70.0 | 72.0 | 72.9 | 69.5 | 71.1 |
> > > | AlphaCLIP + LongGRIT | 69.3 | 82.2 | 82.4 | 93.8 | 94.0 | 74.2 |
> > >
> > > Results show that such a model naturally achieve performance between the original AlphaCLIP and our final PixCLIP, further demonstrating the effectiveness of LongGRIT. Since AlphaCLIP was not trained on long-text data, it performs much worse on image–text retrieval datasets such as ShareGPT4V.
> > >
> > > To further demonstrate the contribution of our dataset, we also conduct data scaling experiments:
> > >
> > > | Data Scale | ImageNet-S @1 | ImageNet-S @5 | DOCCI I2T | DOCCI T2I | RefCOCO Val | FG-OVD trivial |
> > > | --- | ---: | ---: | ---: | ---: | ---: | ---: |
> > > | 20% | 65.7 | 88.5 | 96.7 | 96.0 | 56.8 | 52.6 |
> > > | 50% | 67.5 | 90.0 | 96.5 | 96.3 | 58.4 | 69.2 |
> > > | 100% | 69.6 | 91.2 | 96.7 | 96.4 | 59.9 | 78.9 |
> > >
> > > &nbsp;
> > >
> > > Finally, we sincerely thank you again for your valuable comments, which are very helpful for revising our paper.

---

### Official Review · Reviewer_emZp · 2026-03-09

**Soundness:** 3
**Presentation:** 3
**Significance:** 3
**Originality:** 3
**Overall Recommendation:** 5
**Confidence:** 3

**Summary:**

This paper presents PixCLIP, a CLIP-style vision-language model for any-granularity pixel-text alignment. The goal is to support both mask-conditioned local visual inputs and long, compositional text descriptions in a unified contrastive representation space. To enable this setting, the paper also introduces LongGRIT, a large-scale dataset of roughly 1.5M mask-text pairs constructed with automatic generation and multi-stage verification. On the modeling side, the paper argues that naive mask-long-text contrastive training is unstable, and proposes a three-branch framework including contrastive mask-text alignment, fine-grained cropping alignment, and local-global representation enhancement. The experimental results show strong performance on both fine-grained region-level tasks and standard image-text retrieval benchmarks.

**Compliance With Llm Reviewing Policy:**

Affirmed.

**Final Justification:**

Rebuttal addressed my concerns. I raise score: accept

**Key Questions For Authors:**

(1) The dataset is a major contribution of the paper. Could the authors provide a bit more quantitative analysis of the annotation quality of LongGRIT, for example through a small-scale human evaluation or more explicit error statistics? A stronger answer here would further increase my confidence in the dataset contribution.

(2) The paper argues that naive mask-long-text contrastive training is unstable. Could the authors elaborate a bit more on the failure mode analysis and clarify which branch contributes most under different downstream settings? This would improve the paper’s interpretability, though it would not change my overall positive assessment.

**Limitations:**

Yes

**Strengths And Weaknesses:**

Strengths:
(1) The paper addresses a meaningful gap between existing region-aware CLIP-style methods and long-text retrieval models. Supporting both arbitrary mask inputs and long compositional text in one retrieval-oriented framework is a relevant and timely goal.
(2) The work is comprehensive: it contributes both a new model and a substantial dataset. This makes the paper more impactful.
(3) The three-branch training framework is intuitive and well aligned with the optimization challenges discussed in the paper, especially background leakage and the mismatch between local and global semantics.
(4) The paper is also very timely. It directly compares against LLM2CLIP, the AAAI 2026 best paper, and shows that it can not only further improve image-text retrieval performance, but also equip the model with a new mask-conditioned embedding capability that LLM2CLIP does not target. In that sense, the contribution is not just a stronger set of numbers, but a meaningful extension of what CLIP-style embeddings can represent, which I think is quite inspiring for future work.
(5) The experiments are broad and show that PixCLIP performs strongly on region recognition / retrieval tasks while also remaining competitive on traditional image-text retrieval benchmarks. It compares against both region-focused and long-text-focused baselines, which makes the empirical claims more convincing.
(6) LongGRIT appears to fill a real data gap for learning mask-level alignment with long-form descriptions. I expect this dataset to be useful for future work beyond this paper itself.
(7) I like that the paper is not narrowly optimized for one benchmark only. The problem setting, dataset, and model together point to a broader direction for fine-grained multimodal retrieval and grounding.


Weaknesses
(1) Presentation can still be polished. The paper is generally understandable, but the writing could be tightened in several places. Some parts of the motivation and method description are more verbose than necessary.
(2) Figures and tables could be improved.
The experimental content is strong, but some figures/tables look a bit rough and could be made easier to read.
(3) More discussion of dataset noise would strengthen the paper. Since LongGRIT is automatically constructed, a bit more quantitative discussion of annotation quality or noise robustness would make the dataset contribution even stronger. The current limitation discussion is appreciated, but this aspect could still be expanded.  Overall, I find the paper technically solid, meaningful, and quite compelling. The weaknesses are relatively minor and mostly concern presentation and additional analysis rather than the core contribution.

---

> ### Author Rebuttal · Authors · 2026-03-30
>
> We thank Reviewer `emZp` for the positive and encouraging assessment. We are glad the reviewer found the problem setting meaningful and the model/dataset contribution compelling.
>
> **Concern 1. Some figures and tables look rough and could be easier to read.**
>
> Response: We agree. We will revise the layout and captions of the main figures/tables, improve notation consistency, and add more explanation to the ablation and visualization figures so that the key takeaways are easier to extract. In fact, we have added a significant number of experimental and visualization figures in this revision, which will be included in the final version:
> https://anonymous.4open.science/r/pixclip-figure-BBE0/pixclip-viz-caption.png
>
> **Concern 2. Since LongGRIT is automatically constructed, more quantitative discussion of annotation quality/noise would strengthen the paper.**
>
> Response: We agree, and we will surface the quantitative evidence from Appendix H more prominently in the main text. Specifically, in the first stage of LongGRIT construction, `91.3%` of samples are retained after Qwen-based filtering, and a `2,000`-sample human audit shows over `96%` agreement on correctness and spatial coherence. We agree these numbers should be emphasized more clearly in the current draft.
>
> **Concern 3. Could the authors provide a small-scale human evaluation or more explicit error statistics for LongGRIT?**
>
> Response: Yes. The paper already includes a `2,000`-sample human audit with over `96%` agreement on correctness and spatial coherence; we will move this result into a more visible position and describe more explicitly what is being measured.
>
> **Concern 4. Could the authors elaborate more on the failure modes of naive mask-long-text training and clarify which branch contributes most under different downstream settings?**
>
> Response: Our central claim is that naive mask-long-text contrastive learning is harder than standard image-text training because (i) long captions introduce noisier and denser supervision, and (ii) local representations can drift away from global semantics without explicit coupling. This motivates the three-branch design. Their empirical roles are different, overall, both additional branches improve model performance, with the LG branch demonstrating a stronger improvement.

---

> > ### Author Rebuttal · Reviewer_emZp · 2026-04-01
> >
> > My concerns have been adequately addressed. I raise my score 5: Accept.

---

### Official Review · Reviewer_qiBt · 2026-03-12

**Soundness:** 2
**Presentation:** 2
**Significance:** 3
**Originality:** 3
**Overall Recommendation:** 4
**Confidence:** 3

**Summary:**

The submission studies the problem of aligning image regions with long text. For this purpose, the paper constructed a large-scale region-caption dataset, LongGRIT, with around 1.5M samples annotated by multiple pre-trained VLM.  Based on this dataset, the paper proposed to learn region-caption alignment in the CLIP-style contrastive learning. The model is trained to ensure the consistency between three types of region representations: 1. Cropping the regions from the images as the model’s input; 2. Feed the complete images together with the region masks to the models, and 3. Compute the representation of the complete images and apply average pooling to the patch embeddings falling in the target regions as the region embedding. The resulting model was evaluated and compared on both region-level and image-level classification and retrieval tasks with SOTA solutions, which demonstrate its effectiveness.

**Compliance With Llm Reviewing Policy:**

Affirmed.

**Final Justification:**

The comparison between AlphaCLIP and AlphaCLIP+LongGRIT addresses my concern about the contributions of data and model designs. However, regarding the selection of primary branch, why the pooling branch can not be selected is still confusing to me. Overall, despite some remaining ambiguities, the authors' clarification has convinced me, and I am happy to adjust my rating to a weak accept.

**Key Questions For Authors:**

Please refer to the weakness listed above.

**Limitations:**

Yes

**Strengths And Weaknesses:**

The rationale behind some model designs is unclear and the experiments can be improved substantially.
+ Although the models showed notable performance advantages over existing solutions, the contributions of the data and the model designs are unclear.
+ One interesting observation on Table 6 is that the fine-grained cropping objective brings negligible improvement in pixel-level tasks but helps more on Image-level tasks, especially Urban1k I2T, which doesn’t seem to align with the intention.
+ The proposed method is essentially matching different types of region representations obtained by either cropping, masking, or pooling. However, it is unclear why the vision-language contrastive loss is applied to the masked region embeddings only instead of the remaining two types of region embeddings. The same applied to the region-to-region contrastive learning. Why are the masked region embeddings used as the anchor to match with the other two types?
+ The visualization in Figure 5 is provided with limited discussion, which makes it not so informative.

The paper is somewhat hard to follow, with both formatting issues and missing details.
+ Although an overview of the automated annotation pipeline is provided and described, details like how the VLMs are prompted and how their responses are parsed are missing.
+ Are the notations $\mathcal{P}$ at L269-right and $\rho$ in Eq.(5) defined somewhere?
+ The inference pipeline is also unclear. Are those three types of region representations are all needed or just the masked region embeddings? If all three types of representations are needed, how are they matched with text?
+ formatting issues: incorrect quotation marks (e.g., L205-left,  L216-left, etc) and missing space among words (e.g., L162-right, L166-left, etc).

The problem studied in this paper, to align image regions in arbitrary shapes with long text, is practical. The constructed dataset is potentially beneficial to future studies in this field. However, the proposed model designs need further analysis to decouple their effectiveness from the data, so as to ensure their impact on future works.

---

> ### Author Rebuttal · Authors · 2026-03-31
>
> We thank Reviewer `qiBt` for the careful reading and constructive feedback. Below we respond weaknesses.
>
> **W2:** We understand the concern. ① Regarding “brings negligible improvement in pixel-level tasks”: the LG branch contributes more significantly than the FC branch. After the improvement brought by the LG branch, further gains from the FC branch become limited due to the high difficulty of Ref-SAV retrieval.  ② Regarding “but helps more on image-level tasks, especially Urban1k”: image–text retrieval is not solely determined by structural relationships in the image. The cropping branch reduces irrelevant background and highlights fine-grained local appearance details. This type of sharpening is especially helpful when a long image-level description depends on subtle object attributes.
>
> **W3:** We understand that this concern may arise from a misunderstanding of our training–inference pipeline. This point is straightforward to clarify: First, the masked branch is the ***primary deployment branch***, i.e., the only visual output that we ultimately optimize. During inference, we use only the mask branch, and the resulting embedding is the core representation we aim to optimize. The text branch, LG branch, and FC branch are all designed to align with it during training. The key novelty of our method lies in the fact that this single embedding can support "any-granularity retrieval", without requiring an additional ROI branch as in methods such as FG-CLIP.
>
> As for why the LG and FC branches are aligned to the visual embedding rather than directly to the text features, the reason is intuitive. The final visual embedding we optimize encodes not only appearance information but also positional and interaction cues—note that such information is explicitly present in our training texts. If the FC branch were aligned directly with the text branch, it would be unable to fully match these signals, since cropped images inherently lose such contextual information. In contrast, aligning them to the visual embedding allows the model to preserve both fine-grained details and contextual cues.
>
> We hope this clarifies the rationale behind our design. We also provide ablation studies to support this:
>
> | LG and FC | DOCCI I2T | DOCCI T2I | Urban1K I2T | Urban1K T2I | Flickr30K I2T | Flickr30K T2I |
> | --- | ---: | ---: | ---: | ---: | ---: | ---: |
> | **Align to Text Emb** | 95.1 | 96.3 | 88.8 | 92.9 | 82.4 | 86.3 |
> | **Align to Visual Emb** | **96.7** | **96.4** | **91.0** | **93.7** | **84.2** | **87.0** |
>
> **W4:** We have additionally provided new visualizations, which now clearly illustrate the information we aim to convey, which clearly illustrate the point: https://anonymous.4open.science/r/pixclip-figure-BBE0/pixclip-viz-caption.png
>
> **W5:** Thank you for the suggestion. We will provide the full MLLM details in the appendix of the revised version. A partial preview is shown below:
> For InternVL: What is the object in the image，give me a detailed description.
> For Consistency: I will give you two object descriptions. Please determine whether these two descriptions refer to the same object. If they do, please answer \"Yes.\" If they do not, please explain the reason.\n
>
> **W7:** As addressed in W3, we have explained this above. **In Fig. 6 of the Appendix**, we have already emphasized the difference between training and inference. We hope you can refer to it. Our two additional branches are used only to assist training.

---

> > ### Author Rebuttal · Reviewer_qiBt · 2026-04-01
> >
> > The rebuttal resolves some of my concerns, but there are still several missing/follow-up questions:
> > 1. The individual contribution of the data and model designs is unclear.
> > 2. Why is the branch with masked inputs being considered as the primary branch instead of the ones with pooling or cropping? How will this design choice affect the performance?
> > 3. It is appreciated if the authors can point me to the definition of annotations that I failed to find.

---

> > > ### Author Response · Authors · 2026-04-02
> > >
> > > Thank you for the thoughtful follow-up. We are encouraged that, through your questions, the added evidence from the visualizations and ablation studies has improved the paper. We address your remaining concerns below:
> > >
> > > &nbsp;
> > >
> > > **Q1:** We restate the contributions of this paper.
> > >
> > > Our ultimate goal is **to train a foundation model that can align arbitrary local regions with arbitrarily long text.**
> > >
> > > ① Therefore, at architecture level, we design an additional mask convolution layer on the visual side and replace the text encoder with an LLM. **This is the necessity of the model structure.**
> > >
> > > ② Then, since there is no Region–long-text dataset available for training this objective, we have to design an automated pipeline to construct our own dataset, LongGRIT. **This is the necessity of LongGRIT**, without it, we simply cannot train a model that aligns regions with long text.
> > >
> > > ③ Next, after modifying the model architecture, we directly train using LongGRIT (without any auxiliary branches), and the results are reported in the ablation study Table 6 in our paper. The performance is already competitive, although still inferior to previous SOTA in some aspects. This demonstrates **the effectiveness of the Model Structure + LongGRIT**.
> > >
> > > We further add experiments where AlphaCLIP trained on LongGRIT:
> > >
> > > | Method | ImageNet-S Acc@1 | Flickr30K I2T | Flickr30K T2I | ShareGPT4V I2T | ShareGPT4V T2I |
> > > | --- | ---: | ---: | ---: | ---: | ---: |
> > > | AlphaCLIP | 68.9 | 70.0 | 72.0 | 72.9 | 69.5 |
> > > | AlphaCLIP + LongGRIT | 69.3 | 82.2 | 82.4 | 93.8 | 94.0 |
> > >
> > >
> > > ④ Finally, to address the difficulty of training and the insufficient performance, we design two dedicated branches to assist the features from the main branch during training. The $L_{LG}$ jointly improves global and local capabilities, while $L_{FC}$ enlarges local details as visual anchors, as shown in the ablation study, they all boost performance, show **the effectiveness of the training framework**.
> > >
> > > All components of our paper are organized as above. Each step is designed to achieve the final goal, and has been verified through experiments. The logic is clear and consistent. We hope the reviewer can understand the overall reasoning of our work and no longer feel confused.
> > >
> > > &nbsp;
> > >
> > > **Q2:** It is in fact a straightforward point. The feature source used for training must match that used for inference. Thus, the pooling and cropping branch are used to assist the alignment of the main branch, since only the main branch provides the final visual emb. We explain below why **using the other branch as the model’s main branch, namely the inference branch, is entirely meaningless.**
> > >
> > > Let us consider alternatives.
> > >
> > > **(1).** If the pooling branch were used as the main branch for alignment with text:
> > > ① The model would no longer be end-to-end. For full-image input, one can directly obtain features, but for masked input, an additional pooling step is required to extract region features.
> > > ② Since both training and inference would rely on the pooling branch, the mask emb branch would become meaningless. In our framework, pooling is applied on meaningful full-image features, whereas introducing a separate mask embedding branch would require training a mask conv layer from scratch, increasing training cost and hindering optimization.
> > >
> > > **(2).** If the cropping branch were used as the main branch, the situation is even less reasonable. Directly cropping the image during inference to obtain features discards all positional and relational information. So it can't properly align with textual descriptions involving spatial/relational cues, making the training objective theoretically difficult to optimize.
> > >
> > > **(3).** If the reviewer instead suggests aligning other branches with text during training while still using the Mask emb branch at inference, this would clearly lead to collapse. The feature used during training must match the feature used at inference. Aligning A with B and B with C during training does not imply that A can align with C at inference; feature space mappings are not transitive, is well-known fact in retrieval.
> > >
> > > **(4).** The only theoretically reasonable way is to extend current framework by additionally aligning the other two branches with text, so we did the study:
> > >
> > > | Framework | DOCCI I2T | DOCCI T2I | FG-OVD Hard | ImageNet-S @1 | ImageNet-S @5 | Speed |
> > > | --- | ---: | ---: | ---: | ---: | ---: | ---: |
> > > | Original | **96.7** | **96.4** | **44.2** | **69.6** | **91.2** | **1.0x** |
> > > | + 2branch align-to-text | 96.1 | **96.4** | 40.6 | 67.4 | 90.1 | 0.8x |
> > >
> > > Results show our framework performs better and requires less training cost due to fewer losses.
> > >
> > > &nbsp;
> > >
> > > **Q3:** Thanks for pointing the missing writing, we will correct it.
> > >
> > > Sincerely thank again for your careful review, although you may have some confusion. **If there is anything you want us to further address, please reply to us.**

---

### Decision · Program_Chairs · 2026-04-30

**Decision:**

Accept (regular)

**Comment:**

The paper initially received mixed reviews: WR, WA, WA, R. Reviewers generally recognized the merit of this work, regarding the studied problem meaningful, the proposed training framework intuitive and novel, and the experiments broad and competitive. There are also some concerns and suggestions about 1) model design justification (qiBt), 2) writing clarity (qiBt, emZp, BXGd, QZhF), 3) more analysis about the dataset and experiments (emZp, QZhF), and 4) technical novelty (QZhF).

The rebuttal was overall persuasive. After the rebuttal, Reviewer qiBt increased the rating from WR to WA, turning positive about this work. Both reviewer emZp and BXGd increased their ratings from WA to A, being more supportive to this work. Reviewer QZhF still had concerns about the novelty and the dataset open-source issue. The authors have confirmed to open source the LongGRIT dataset in the discussion with BXGd. So this won’t be a concern. Regarding the novelty concern, Reviewer BXGd provided additional comments in the discussion among reviewers to support the novelty of the proposed three-branch design.

The AC checked the paper, rebuttal, and all review comments, and overall regards the merit of this work outweighing its weakness. Thus, the AC recommends accepting the paper.